# Humidity response in *Drosophila* olfactory sensory neurons requires the mechanosensitive channel TMEM63

Songling Li[1,2], Bingxue Li[1,2], Li Gao[1], Jingwen Wang [1] & Zhiqiang Yan [1,2✉]

Birds, reptiles and insects have the ability to discriminate humidity levels that influence their survival and geographic distribution. Insects are particularly susceptible to humidity changes due to high surface area to volume ratios, but it remains unclear how humidity sensors transduce humidity signals. Here we identified Or42b-expressing olfactory sensory neurons, which are required for moisture attraction in *Drosophila*. The sensilla housing Or42b neurons show cuticular deformations upon moist air stimuli, indicating a conversion of humidity into mechanical force. Accordingly, we found Or42b neurons directly respond to humidity changes and rely on the mechanosensitive ion channel TMEM63 to mediate humidity sensing (hygrosensation). Expressing human TMEM63B in *Tmem63* mutant flies rescued their defective phenotype in moisture attraction, demonstrating functional conservation. Thus, our results reveal a role of *Tmem63* in hygrosensation and support the strategy to detect humidity by transforming it into a mechanical stimulus, which is unique in sensory transduction.

---

[1] State Key Laboratory of Medical Neurobiology and MOE Frontiers Center for Brain Science, Ministry of Education Key Laboratory of Contemporary Anthropology, School of Life Sciences, Fudan University, Shanghai 200438, China. [2] Institute of Molecular Physiology, Shenzhen Bay Laboratory, Shenzhen 518132, China. ✉email: zqyan@szbl.ac.cn

Humidity provides a cue for terrestrial animals to migrate toward favorable environments, which ensure their survival and reproduction[1–6]. Notably, insects have the ability to extract information about their surroundings via humidity sensation. For instance, the hawkmoth *Hyles lineata* uses humidity levels to assess the nectar availability of blooming flowers that produce a higher level of humidity compared with their ambiance[7]. In mosquitoes, the disease vectors, moisture serves as a key attractant for host seeking[8,9]. *Drosophila* are equipped with sophisticated hygrosensory organs to detect moisture levels and possess various humidity-induced behaviors[10–19]. Water-sated flies are averse to moisture, while dehydrated flies show attraction to moisture as well as moisture searching behavior[11–18]. Besides, multiple *Drosophila* species show diverse humidity preferences and have different distributions, for example, rainforest flies and desert-dwelling flies prefer 85% and 20% relative humidity (RH), respectively[6,12]. Although recent studies have revealed the neural and molecular basis underlying humidity preference in *Drosophila*[12–15], the mechanisms by which humidity stimuli are transduced into electrical signals and guide moisture attraction behavior remain largely obscure.

Humidity signals are transduced and encoded by hygroreceptors; however, the identification of molecular receptors in hygrosensory neurons has been hampered by the elusive nature of humidity, which is the amount of water vapor in the atmosphere. Three concepts that explain how humidity changes activate a hygroreceptor have been proposed[6,20–22]. In the mechanical hygrometer model, change of humidity levels causes shape change of sensilla structures so that mechanosensitive molecules are activated to mediate hygrosensation. In the osmometer model, humidity-dependent osmolality change occurs in the sensilla lymph and evokes hygrosensation. Additionally, the psychrometer model posits that thermosensitive molecules function as humidity transducers, which detect the evaporation cooling from the sensilla lymph. These concepts are not mutually exclusive as sensory inputs from different pathways may act in concert to mediate hygrosensation by a multisensory integration mechanism. Previous studies have established several ionotropic receptors (IRs) including IR40a, IR68a, IR93a and IR25a as attractive candidates for humidity receptors[12–15], but it remains unclear whether these molecules respond directly to mechanical or thermal stimulus. Thus, molecular evidence for those concepts regarding humidity transduction mechanisms has been lacking.

In this study, we identified a mechanosensitive ion channel TMEM63, which functions in a group of olfactory receptor (OR)-expressing neurons to mediate hygrosensation. Using a newly developed assay, we first identified a group of olfactory sensory neurons (OSNs), the Or42b neurons, which work in synergy with IR68a-expressing moist cells to guide moisture attraction behavior. The sensilla that house the dendritic branches of Or42b neurons show shape change upon increasing humidity, which suggests humidity stimulus can be converted into mechanical force on the dendritic membrane. We further show that the humidity response of Or42b neurons depends on TMEM63, a member of the recently identified family of mechanosensitive channels, thus providing molecular evidence for the model in which a mechanosensory pathway contributes to humidity sensing. The physiological function of TMEM63 proteins in the animal kingdom is largely unknown, we found human TMEM63B rescues moisture attraction defects in *Tmem63* mutant flies, demonstrating functional conservation in hygrosensation.

## Results

**The Or42b neurons mediate 70% RH induced attraction with IR68a neurons.** To study the neural circuits and molecular mechanism of moisture attraction behavior, we first sought out to establish an experimental paradigm (Fig. 1a) adapted from previous hygrotaxis assays[18,19]. Briefly, flies were placed in a Petri dish that was covered with nylon net. After desiccation for 6 h, the dish was placed above a 24-well plate with 2 holes filled with super-saturated salt solutions[23] immediately beneath the nylon net, creating a humidity gradient between 20% RH and 70% RH. The attraction index was then calculated as the percentage of flies in the region of higher humidity every 10 seconds. The chance level of attraction to the 70% RH region is estimated to be ~12% based on the proportion of the moist area in a random distribution. The humidity gradient was stable during the 120 s experimental period (Supplementary Fig. 1a). Wild type flies showed robust moisture attraction within 50 s, with 61.9 ± 1.4 % of flies attracted to 70% RH during the plateau stage (Fig. 1b, c and Supplementary Fig. 1e and Supplementary Movie 1). The attraction index was independent of the group density of animals that were introduced to the test (Supplementary Fig. 1c). Neither water sated flies nor those starved overnight showed a moisture attraction (Supplementary Fig. 1d), reflecting an essential role of internal state in driving moisture seeking, consistent with previous results[16,17]. When changing the humidity setting to a 70%-96% RH gradient, wild type flies showed an increased attraction, with 78.3 ± 1.8% flies in the 96% RH region (Supplementary Fig. 2a–d). As single experiment can be completed in about 2 min, this time-saving assay facilitates the repetition of results and is suited for functional screens.

Antennal segments have been implicated in humidity discrimination[11,18], we therefore asked whether antennal sensory neurons are indispensable for moisture attraction. We found that surgically removing the 3rd antennal segment completely disrupted humidity discrimination between 20 and 70% RH, with only 15.3 ± 1.4 % of flies distributed in the moist region (Fig. 1b, c and Supplementary Movie 2), in accord with previous studies[11,18]. Similar results were yielded in flies tested with a 70%-96% RH gradient (Supplementary Fig. 2b-d). The attraction index of antenna ablated animals was close to the index of the uniform distribution which was approximately 12%, suggesting that hygroreceptors in the antenna are essential for the humidity-induced attraction in flies under desiccation stress.

Four antennal cell groups have been associated with hygrosensation, including dry cells and moist cells located near the sacculus[12–15], some neurons in the coeloconic sensilla and a group of neurons targeting the basiconic sensilla[10,11]. To identify the cellular substrates underlying the observed moisture attraction behavior, we performed a screen by crossing Gal4 drivers for these cells with *UAS-ReaperHid*[24–26], which gives rise to programmed cell death. The efficiency of neuronal ablation is nearly 100%, as verified by checking fluorescence signals from related cells (Supplementary Fig. 3a). We found neuronal ablation driven by two lines, *IR68a-Gal4* and *Nan-Gal4*, led to obvious defect in the attraction to 70% RH when compared with the Gal4 control group (Fig. 1g). *IR68a-Gal4* specifically labels moist cells which mediate moisture sensing in an IR-dependent manner. However, *Nan-Gal4* labeling is not confined to the third antennal segment. To find the neuron group contributing to moisture attraction, we examined the axon projections of *Nan-Gal4*-expressing neurons in the antennal lobe. Multiple glomeruli showed robust GFP staining, one of them appears to be the DM1 glomerulus, which was confirmed by the results that an *Or42b-Gal80* line fully erased this labeling by *Nan-Gal4* (Supplementary Fig. 3b). Strikingly, expressing Reaper and Hid under the control of *Or42b-Gal4* line impaired the attraction to moisture, with only 35.9 ± 1.6% of flies staying at 70% RH (Fig. 1d, e). Similar results were obtained in water-deprived flies carrying *Or42b-Gal4* and *UAS-Kir2.1*, an inward-rectifying potassium channel that blocks

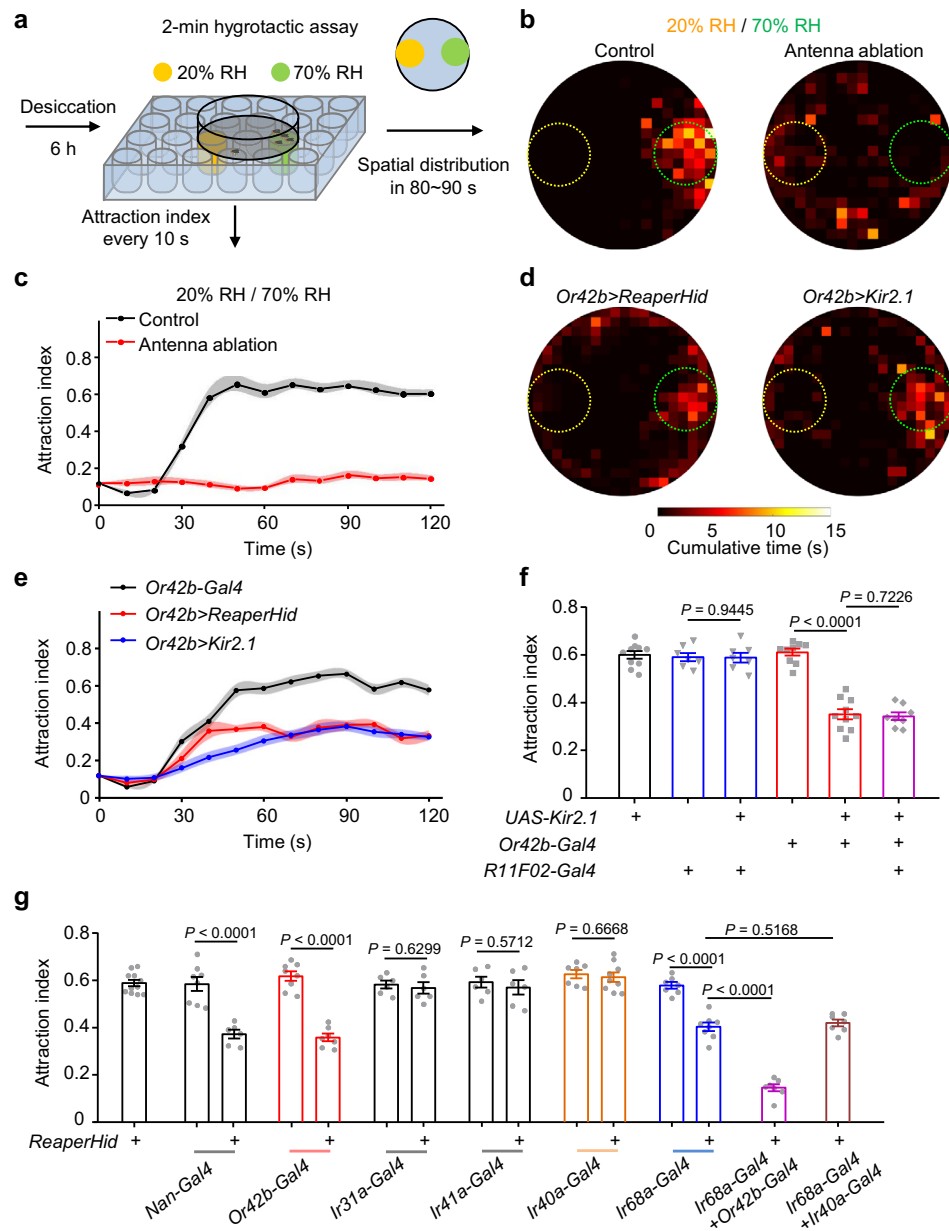

**Fig. 1 Humidity-guided attraction to 70% RH requires Or42b OSNs. a** Schematic diagram of the humidity-induced attraction behavioral assay. The yellow region represents 20% RH generated by saturated LiCl solution, and the green region denotes 70% RH produced by saturated NaCl solution. A total of 20–40 flies were used per assay. **b** Spatial distribution of control flies (left) and flies with the antennae removed (right) assayed in a 20% to 70% RH gradient. $w^{1118}$ flies were used as control flies. Control, $n = 22$ flies; Antenna ablation, $n = 20$. The yellow dashed circle denotes the area above 20% RH, and the green circle indicates the 70% RH region. Color intensity represents the cumulative time the flies spent on each pixel during 80–90 s after the onset of the assay. **c** The time course indicating the attraction indexes of control and antenna-ablated flies. Control, $n = 14$ assays; Antenna ablation, $n = 10$. **d** Spatial distribution of Or42b > ReaperHid (left) and Or42b > Kir2.1 (right) flies assayed under the 20% to 70% RH condition. Or42b > ReaperHid, $n = 24$ flies; Or42b > Kir2.1, $n = 25$. **e** The time course indicating the attraction indexes of flies carrying indicated transgenes. Or42b-Gal4, $n = 12$ assays; Or42b > ReaperHid, $n = 7$; Or42b > Kir2.1, $n = 11$. **f** Attraction indexes after inhibiting thermosensory neurons or Or42b-expressing neurons or both. $n = 10, 7, 7, 9, 10, 8$ assays. **g** Humidity-guided attraction to 70% RH in flies with removal of different neuron groups by crossing UAS-ReaperHid with the indicated Gal4 lines. $n = 11, 8, 6, 8, 7, 6, 6, 6, 7, 9, 7, 8, 7, 7$ assays. Two-tailed unpaired t test for two groups, one-way ANOVA followed by Holm-Sidak's test for multiple comparison. For (**c**), (**e**), data points are mean values and shaded area represents ± SEM. For (**f**), (**g**), data are mean ± SEM. Source data are provided as a Source Data file.

neuron function (Fig. 1d–f). Or42b-Gal4 labels a single group of OSNs innervating large basiconic sensilla (ab1 sensilla)[27–31]. This group of sensory neurons have been reported to be essential for innate attraction to vinegar and odor-induced food searching[32,33]. These studies together with our results indicate Or42b neurons play pivotal roles in detecting food and water, two essential factors for animals' survival.

We noted that silencing or ablating Or42b neurons alone did not lead to a complete loss of 70% RH-guided attraction (Fig. 1d–f). The residual attraction to 70% RH may suggest a multisensory integration mechanism for this behavior, since humidity sensation has been shown to involve the action of thermosensory pathway[5,34]. Although no temperature gradient was presented in our behavior assay (Supplementary Fig. 1b),

evaporative cooling can occur and provide thermal cues that affect animal behavior when flies are moving across a humidity gradient. To test this possibility, we assayed the *R11F02-Gal4* line, which was recently reported to label all the six thermosensory neurons[35]. When silencing all the thermosensory neurons in the arista, the moisture attraction in 20-70% RH gradient remained intact (Fig. 1f). Then we concurrently blocked the thermosensory neurons and Or42b neurons, the residual moisture attraction still persisted in these flies (Fig. 1f).

Due to the fact that ablating IR68a neurons also caused a partially impaired phenotype, we propose that IR68a neurons may work in parallel with Or42b neurons to guide moisture attraction behavior. Consistent with this idea, removal of both the two neuron groups nearly eliminated the attraction to 70% RH, with 14.9 ± 1.5 % of flies locomoting in the moist region (Fig. 1g). Moreover, we showed that attraction to 96% RH was mediated by the dry cells and the moist cells in the sacculus sensilla but was independent of Or42b neuron function (Supplementary Fig. 2e), which coincides with previous results[15]. In conclusion, these findings suggest that flies rely on different neuron groups to navigate in different humidity range (Supplementary Fig. 4), and that Or42b neurons are the humidity sensors selectively required for the 70% RH -induced attraction.

**Humidity-dependent shape change of ab1 sensilla**. We then investigated the mechanism by which Or42b neurons regulate animal behavior in the 20-70% RH gradient. Among existing models for the mechanism of hygrosensation[6,20–22], humidity changes are transformed into mechanical or thermal cues before reaching the hygroreceptors. We thus tested if the basiconic sensilla labeled by *Or42b-Gal4*-driven GFP possess specific physical properties to suit the hygrosensory function. We first set up a humidity stimulation system in which we realized rapid humidity changes from 46.64 ± 0.39% RH to 60.88 ± 0.42% RH by applying airflow with different humidity levels (Fig. 2a). When switching from dry air to humid air, the ab1 sensilla tended to straighten, which might result from the hygroscopically induced swelling of the cuticular wall. To quantify this shape change, the sensilla hair was fitted with a curve and the average radius of curvature was measured. Intriguingly, upon exposure to a humidity increase, the radius of curvature for ab1 sensilla increased from 21.73 ± 1.37 μm to 30.94 ± 1.56 μm (Fig. 2b, c, f), which is indicative of a membrane tension change in the sensory endings of Or42b neurons. By contrast, switches between dry airflows had no such effects, indicating the mechanical stimulus from airflow itself was unable to elicit sensilla deformation (Fig. 2d–f). Moreover, converting moist air back to dry air returned the change in curvature. The curvature change happened within 1 s in response to humidity change, and can last throughout the the entire period of moist airflow (Supplementary Fig. 5f). Considering the similar material constituting the cuticular wall that may interact with water molecules, the humidity-induced deformation might occur likewise in other basiconic sensilla. We examined the GFP negative basiconic sensilla and found they also showed shape change upon exposure to humidity stimulation (Supplementary Fig. 5a–e). A detailed analysis revealed a negative linear correlation between the initial radius of curvature and cuticular deformation ratio (Fig. 2g, Pearson's correlation, $r = −0.7211$, $p < 0.001$), which suggests the curved sensilla are more responsive to humidity while straight sensilla may react poorly.

***Tmem63* drives 70% relative humidity-induced attraction**. We therefore hypothesized that the hygrosensory transduction molecule in Or42b neurons might be a mechanosensitive channel.

To test this hypothesis, we screened a number of mutants and RNAi lines disrupting ion channel genes involved in mechanotransduction including Piezo, Tmc and TRP channels Inactive, Nanchung and NompC (Fig. 3a and Supplementary Fig. 6a). Since *nanchung* has been reported to play essential roles in the thirst state sensor in central brain[36], which can also regulate the moisture attraction behavior, we chose to perform RNAi-mediated knockdown of this gene in Or42b neurons. All of these tested lines behaved similarly to the control. Although *inactive* and *nanchung* were both required for the humidity choice between 0% and 100% RH in water sated flies[11], our study excluded their roles in the 70% RH-induced attraction in desiccated flies. The differences in humidity settings and internal state of animals might lead to distinct results. In addition, neither a P-element insertion mutant for *Or42b* (*Or42b^{EY}*) nor *Orco^2* mutants exhibited behavioral defects in humidity-guided attraction to 70% RH (Fig. 3a), indicating that ORs in Or42b neurons are not involved in hygroreception.

We also considered mechanosensitive channels for which the physiological function has not been well elucidated. *Tmem63* is a member of the newly identified family of mechanosensitive ion channels[37,38]. Using nonpermeabilized staining of the myc-tag inserted in the N-terminal region of *Dm*TMEM63, we showed the surface expression of TMEM63 in *Drosophila* S2 cells (Supplementary Fig. 7a). Subsequently, we observed high threshold stretch-activated currents (with a $P_{50}$ of $−83.25 ± 5.37$ mmHg) in S2 cells transfected with *Dm*TMEM63-GFP (Supplementary Fig. 7b–e), which is consistent with prior study[38]. Thus, like its homologs in other eukaryotic species[37–40], *Dm*TMEM63 displays current responses to mechanical stimuli. To determine whether the *Tmem63* gene is involved in moisture attraction, we generated a *Tmem63* null mutant allele (*Tmem63^{KO}*) (Supplementary Fig. 8a), in which nearly half of the coding sequence was replaced via homologous recombination facilitated by CRISPR/Cas9 system[41]. This deletion resulted in an undetectable level of the *Tmem63* mRNA transcript in *Tmem63^{KO}* flies, which was confirmed by RT-PCR (Supplementary Fig. 8b). The *Tmem63^{KO}* animals were viable and did not exhibit gross defects in coordination (Supplementary Fig. 8c). When introducing the *Tmem63^{KO}* flies into the 20-70% RH gradient, we observed an evident decrease in the percentage of flies attracted to moist region (Fig. 3a). Similar to Or42b neuron ablated flies, *Tmem63^{KO}* mutants were normally attracted to moisture in the 70-96% RH gradient (Supplementary Fig. 2d). These data suggest that *Tmem63* is specifically required for 70% RH-induced attraction.

**TMEM63 is expressed in multiple humidity responsive neurons**. The involvement of *Tmem63* in moisture attraction raises the question of whether TMEM63 is expressed in neurons responsible for hygrotaxis behavior. By generating a knock-in reporter line *Tmem63^{LexA}* (Supplementary Fig. 8a), we found that all neurons marked by *Or42b-Gal4*-driven RFP were recognized in *Tmem63^{LexA}*-labeled cells (Supplementary Fig. 9a). Accordingly, overexpression of Gal80 under the control of the *Tmem63^{LexA}* driver fully inhibited the labeling of *Or42b-Gal4* in OSNs (Supplementary Fig. 9b). In the antennal lobe, *Tmem63^{LexA}*-labeled neuron projections also resided in glomerulus DM1, the region innervated by *Or42b-Gal4*-labeled neurons (Fig. 3b). Although we were not able to examine the expression profile of TMEM63 in each of the ~50 OSN groups, we found that TMEM63 is expressed in both OR-expressing cells and IR-expressing neuron groups (Supplementary Fig. 10a). By double-labeling experiments, we further showed that most cells marked by *Ir40a-Gal4* were co-localized with TMEM63 positive cells

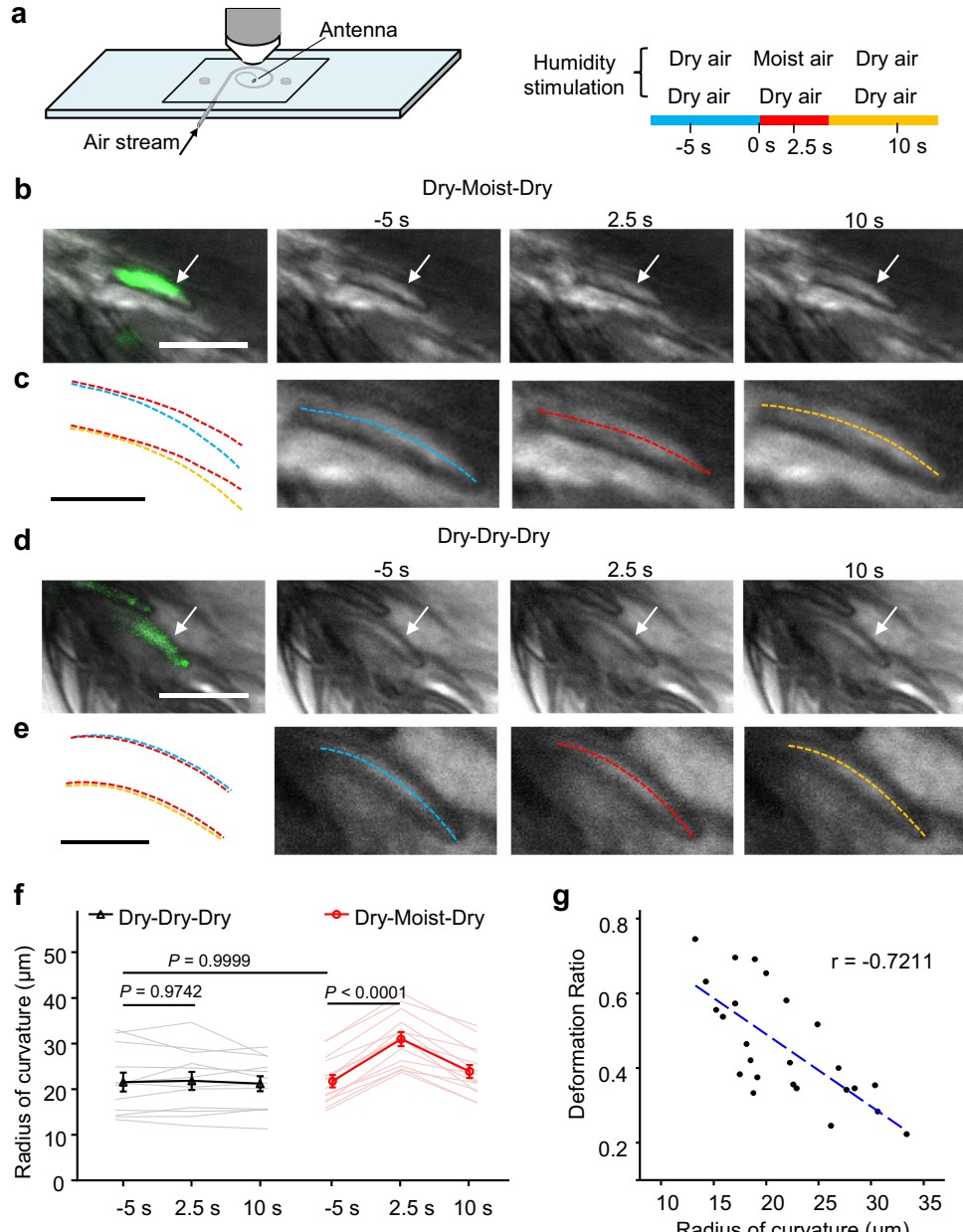

**Fig. 2 The deformation of sensilla housing Or42b neurons upon humidity change. a** Schematic depicting the experimental set up for sensilla imaging, the two humidity stimulations applied was indicated on the right. **b** Time-lapse images showing curvature changes of ab1 sensilla when challenged with sequentially applied dry-moist-dry airflows, the ab1 sensillum labeled by CD4-tdGFP is indicated by a white arrow. Scale bar, 10 μm. **c** Zoomed in images of the GFP positive sensillum indicated in (**b**), dashed lines represent the curves fitted to the shape of the sensillum. Scale bar, 5 μm. **d** Images showing morphology of ab1 sensilla when challenged with switches of dry airflows without altering the humidity. Scale bar, 10 μm. **e** Zoomed in images of the GFP positive sensillum indicated in (**d**). Scale bar, 5 μm. **b**–**e** Representative images from over three independent replicated experiments. **f** Summary of curvature change of GFP positive sensilla under dry-dry-dry or dry-moist-dry air stimuli. $n = 12$ and 14 sensilla. Data are mean ± SEM., two-way ANOVA followed by Sidak's post hoc test. Genotype of flies used for (**b**–**f**) was *Or42b-Gal4/UAS-CD4-tdGFP*. **g** Scatter plot of deformation ratio (increase rate of radius of curvature) versus the initial radius of curvature for all 25 large basiconic sensilla (both GFP positive and GFP negative) that responded to moist air. $n = 25$ sensilla. Source data are provided as a Source Data file.

except for a few *IR40a-Gal4*-expressing neurons ($9.0 \pm 0.3$) located near the sacculus chamber II (Supplementary Fig. 10b, d). When *Tmem63[LexA]* was combined with *IR68a-Gal4*, the driver for moist cells, we observed no overlap between the two neuronal groups (Supplementary Fig. 10c, e).

The broad expression pattern of TMEM63 in the antenna raises the need to revisit the humidity sensitivity of OSNs that express TMEM63. To address this problem, we used the knock-in *Tmem63[LexA]* driver to conduct calcium imaging of the antennal

lobe[13,42] in live flies expressing GCaMP6m, a genetically encoded calcium sensor (Fig. 3c, d). We observed that a conversion from dry airflow to moist airflow led to robust increases in GCaMP fluorescence in the axon termini targeting three regions, the DM1, DL2 and DC4 glomeruli (Fig. 3f, j), suggesting that Or42b (DM1-targeting) and IR75abc (DL2-targeting) expressing OSNs and a subset of IR64a (DC4-targeting) positive OSNs are moisture-activated humidity sensors. The opposite humidity change activated the VP4 glomerulus innervated by IR40a-

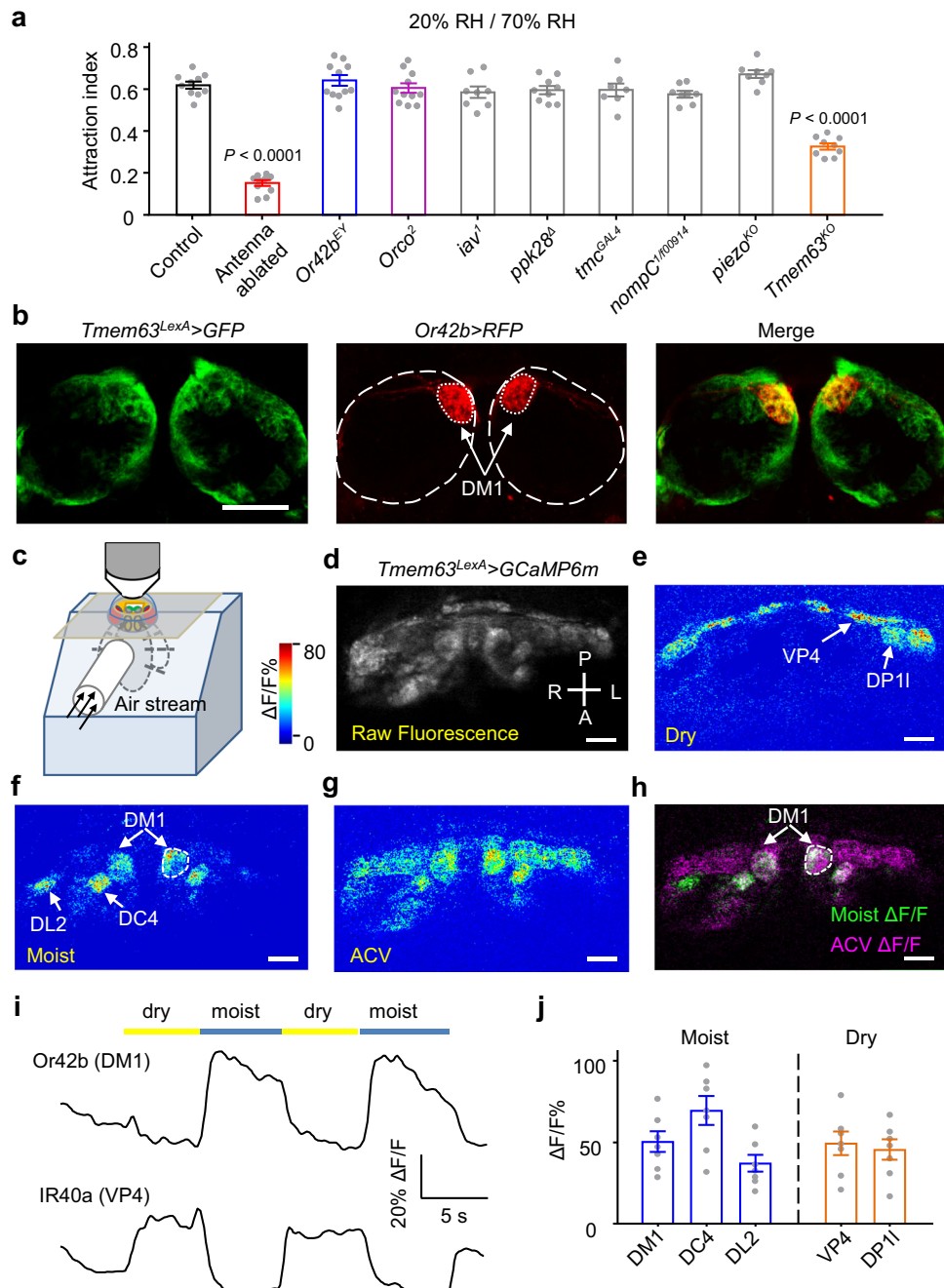

**Fig. 3 Identification of humidity sensitive OSNs by using a knock-in driver of *Tmem63*. a** Plateau attraction indexes for several olfactory receptor and mechanosensitive channel mutants tested in the 20–70% RH condition. $n = 10, 10, 11, 11, 8, 9, 7, 8, 8, 9$ assays. One-way ANOVA followed by Dunnett's test for multiple comparison with the control group. **b** *Tmem63 > GFP*-labeled and *Or42b > RFP*-labeled axons project to overlapping regions of the antennal lobe. White dashed lines indicate the border of DM1 glomerulus and antennal lobe. Scale bar, 50 μm. Genotype: *UAS-mCD8-RFP,LexAop2-mCD8-GFP; Tmem63^{LexA}/+; Or42b-Gal4/+*. Representative images of three biological replicates. **c** Schematic illustration of in vivo calcium imaging in the antennal lobe. **d** Raw fluorescence image of *Tmem63*-expressing axons (in a *Tmem63^{LexA}/+; LexAop2-GCaMP6m/+* animal) innervating the antennal lobe. Scale bar, 20 μm. **e–g** Pseudo color images showing GCaMP6m responses ($\Delta F/F_O$) of *Tmem63*-expressing neurons to humidity stimulations or 1% apple cider vinegar (ACV). Scale bar, 20 μm. **h** $\Delta F/F_O$ response to moist air (green) overlaid on $\Delta F/F_O$ response to vinegar (magenta). Scale bar, 20 μm. **d–h** Representative images from over three independent replicated experiments. **i** Representative traces for moisture-activated response of DM1 glomerulus (upper) and dry air-activated response of VP4 glomerulus (lower) in different humidity changing cycles. **j** Peak $\Delta F/F_O$ in response to humidity changes. $n = 7$ flies for each group. Data are mean ± SEM. Source data are provided as a Source Data file.

expressing dry cells (Fig. 3e, i, j), which is in accord with previous results[12,13]. The DP1l glomerulus, where IR75a positive neuron projections reside, was also identified as an additional dryness responding unit in our experiment (Fig. 3e, j). By comparing the activation patterns of humidity response and vinegar response, we

further confirmed the identity of these glomeruli (Fig. 3g, h). The stimulation protocol used here generated humidity changes between 24.66 ± 0.62% RH and 66.67 ± 0.19% RH, which was rarely controlled in prior studies involving a water stimulus. This probably explains why previous work has not identified a role of

Or42b neurons in humidity sensing[43,44]. To probe the role of each identified humidity sensor in mediating the *Tmem63*-dependent behavioral phenotype, we knocked down *Tmem63* in these OSNs in turn and analyzed how they affect the attraction behavior. However, we only found a deficit in 70% RH-guided attraction following *Or42b-Gal4* specific knockdown of *Tmem63* (Supplementary Fig. 6b), suggesting that *Tmem63* contributes to moisture attraction by functioning in Or42b neurons.

**Humidity transduction in Or42b neurons requires TMEM63s.** To explore the endogenous localization of TMEM63 in antennal neurons, we generated another knock-in reporter line *Tmem63^EGFP-Gal4* with an EGFP tag in-frame fused at the C-terminus of TMEM63 (Supplementary Fig. 8a). Immunofluorescence of GFP was detected in the sensory cilia of Or42b neurons (Fig. 4a and Supplementary Fig. 9c), where sensory transduction occurs. These results raise the possibility that TMEM63 might function as a humidity receptor.

We next analyzed in detail the behavioral deficiency of *Tmem63* mutant flies. The behavior pattern of *Tmem63^KO* flies was distinct from that of wild type flies. The distribution of wild type flies was gradually restricted to the region over the hole generating 70% RH (Fig. 4b, e). In contrast, *Tmem63^KO* flies failed to gather like wild type flies when exposed to the same humidity gradient setting, with only 33.6 ± 1.5 % of mutant flies attracted to 70% RH (Fig. 4b, c). We also found that the *Tmem63^KO* mutant flies locomoted within a larger area beyond the 70% RH hole (Fig. 4e), while they showed a locomotion speed comparable to that of wild type flies under the humidity conditions in which the flies were raised (Supplementary Fig. 8d). To exclude the possibility that the observed behavior phenotype is caused by genetic background effects, we analyzed *Tmem63^KO* in trans with a deficiency that harbors a deletion spanning the *Tmem63* gene. The defect in the trans-heterozygous flies was comparable to the *Tmem63^KO* homozygote phenotype (Fig. 4f). In addition, the abnormality of the *Tmem63^KO* mutants in humidity-induced attraction behavior was fully restored by expressing *Drosophila* TMEM63 under control of *Or42b-Gal4* (Fig. 4d, g). As hygrotaxis in 70–96% RH gradient is independent of *Tmem63* and the attraction index in this humidity range has been reported to rely on the degree of water loss[16,45], the attraction to 96% RH can reflect the thirsty state of *Tmem63^KO* mutant and rescued flies. Taking advantage of this method, we confirmed that mutant and rescued flies were similarly desiccated in our behavioral assay, in that the attraction indices for 96% RH in these flies showed no significant difference from wild type flies (Supplementary Fig. 2d and Supplementary Fig. 11a). Taken together, our results demonstrate that *Tmem63* functions in Or42b neurons, autonomously regulating humidity-guided attraction to 70% RH.

Given the complementary role of IR68a neurons in the remaining moisture attraction when Or42b neurons were ablated, we tested the loss-of-function mutations in IRs. The attraction indices of *IR25a^2* and *IR68a^MB05565* mutant flies were reduced (albeit not abolished) to similar levels when tested in a 20% to 70% RH gradient (Fig. 4f), which is in line with the screening results showing that removal of all sacculus humidity sensors was insufficient to eliminate the 70% RH-induced attraction (Fig. 1g). Double mutants carrying *Tmem63^KO* and *IR68a^MB05565* showed a disrupted moisture attraction, with 13.9 ± 1.2% of flies staying in the 70% RH (Fig. 4f), which is close to the level of random distribution (~12%). When changing the humidity setting to a 70% to 96% RH gradient, *IR25a^2* flies showed a humidity-blind phenotype (Supplementary Fig. 2d), in accord with the severe defect reported previously[12,13,15].

Since mammalian TMEM63s share homology with *Drosophila* TMEM63, and more importantly, the in vivo function of mammalian *Tmem63* genes has just begun to be revealed[46–48], we asked whether human TMEM63s could rescue the defective phenotype of *Tmem63^KO* mutants. To answer this question, we expressed human *Tmem63* genes in the *Or42b-Gal4*-labeled neurons of *Tmem63^KO* mutants. We found that human TMEM63B but not TMEM63A or TMEM63C, fully restored the defect of *Tmem63^KO* mutant flies in the moisture attraction behavior (Fig. 4b, d, e, g and Supplementary Fig. 11b). Hence, human TMEM63B appears to recapitulate the role of *Drosophila* TMEM63 in sensory neurons.

We then analyzed the function of *Tmem63* in the humidity sensitivity of Or42b neurons. In contrast to control flies, the *Tmem63^KO* mutants showed dramatically reduced calcium responses to humidity changes, which were restored by expressing *Drosophila Tmem63* (*UAS-DmTMEM63*) in these neurons (Fig. 5a–c). By contrast, when exposed to odors in low concentration, the Or42b neurons of the *Tmem63^KO* mutants showed similar calcium responses compared to those of wild type animals (Fig. 5e–g). When challenged with higher concentrations of vinegar, the calcium responses of Or42b neurons in *Tmem63^KO* mutants were still indistinguishable from controls (Fig. 5f). These results indicate that TMEM63 is dispensable for the detection of vinegar and ethyl propionate. Furthermore, to validate that the Or42b neurons are the driver of DM1 glomerulus activity, we showed that surgically removing the 3rd antennal segment nearly abolished calcium activity in glomerulus DM1 (Fig. 5d). We also found the calcium response to dry stimulation in VP4 glomerulus and the moisture sensitivity of DC4 glomerulus are independent of *Tmem63* (Supplementary Figs. 12 and 13). Our calcium imaging data suggest that TMEM63 is selectively required for the hygrosensory transduction in Or42b neurons. The differences in the requirement of TMEM63 in the humidity response in distinct neuron populations might result from the morphological differences in the sensilla that house them.

## Discussion

In existing concepts explaining the mechanism of hygrosensory transduction[6,20–22], humidity receptors are either mechanosensitive or thermosensitive molecules. Previous studies have revealed the essential roles of IR40a, IR68a, IR93a and IR25a in hygrosensation[12–15], but due to the difficulty of function analysis of theses molecules in heterologous expression system, how they respond to humidity still need further investigation[6]. By neuron ablation methods and mutant analysis, we revealed a mechanosensory pathway that specifically contributes to 70% RH-induced attraction. Our data provide structural and molecular evidence supporting the concept that humidity changes can be transformed into mechanical cues which evoke hygrosensory inputs via mechanosensitive molecules. We found that moisture changes the cuticular curvature of ab1 sensilla, which may alter the membrane tension of associated sensory cilia expressing the mechanosensitive channel TMEM63. Previous in vitro studies have shown that TMEM63/OSCAs are sensitive to both osmotic stress and negative pressure[37,38], the present study shows that *Drosophila* TMEM63 confers stretch-activated currents when transfected into S2 cells. Thus *Dm*TMEM63 might be a potential primary molecular receptor that can sense the membrane deformation resulting from humidity changes (Supplementary Fig. 14), although it could not be ruled out that TMEM63 could mediate humidity perception by sensing osmotic pressure changes. Intriguingly, our study reveals an evolutionary strategy to detect the obscure humidity or water vapor by converting it into a defined mechanical deformation, which is unique in the sensory system.

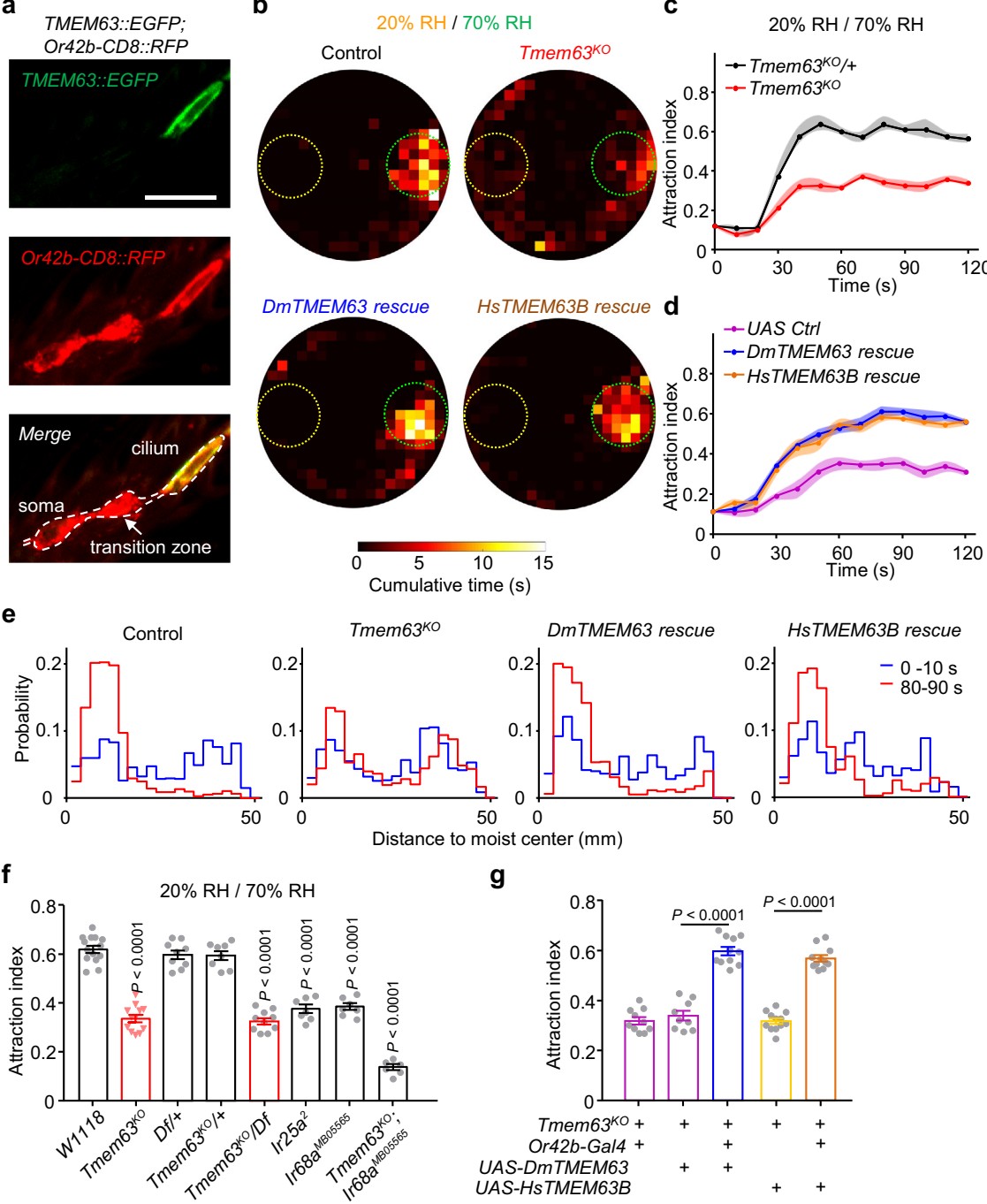

**Fig. 4 *Tmem63* functions in Or42b neurons to mediate humidity-induced attraction to 70% RH. a** Immunostaining of TMEM63::GFP (anti-GFP) labels the sensory cilia of Or42b neurons. White dashed line represents the border of a single Or42b neuron, with the main anatomical features shown. Each result was repeated three times. Scale bar, 10 μm. **b** Spatial distribution of control (upper left), *Tmem63^{KO}* (upper right), *DmTMEM63-Rescue* (lower left) and *HsTMEM63B-Rescue* (lower right) flies assayed in a 20% to 70% RH gradient. The yellow dashed circle denotes the area above 20% RH, and the green circle indicates the 70% RH region. $n = 25$ flies for each group. **c** The time course indicating the attraction indexes of heterozygous and homozygous *Tmem63^{KO}* flies assayed in the arena with a 20% to 70% RH setting. *Tmem63^{KO}/+*, n = 8 assays; *Tmem63^{KO}*, n = 12. **d** The time course indicating the attraction indexes of *UAS Ctrl*, *DmTMEM63-Rescue* and *HsTMEM63B-Rescue* flies assayed in the arena with a 20% to 70% RH setting. $n = 9, 11, 11$ assays. Genotypes are *UAS Ctrl*: *Tmem63^{KO}; UAS-DmTMEM63/+*, *DmTMEM63-Rescue*: *Tmem63^{KO}; UAS-DmTMEM63/Or42b-Gal4* and *HsTMEM63B-Rescue*: *Tmem63^{KO}; UAS-HsTMEM63B/Or42b-Gal4*. **e** Probability distributions of distance between flies and moist center during 0–10 s (blue) and 80–90 s (red) after the onset of the assay. $n = 25, 25, 24, 25$ flies. Each result was reproducible in three independent experiments. **f** Attraction indexes of flies with indicated genotype tested in the 20–70% RH condition. $n = 14, 12, 9, 8, 10, 7, 7, 6$ assays. One-way ANOVA followed by Dunnett's test for multiple comparison with the control group. **g** Expression of *Drosophila* TMEM63 or human TMEM63B in *Or42b-Gal4* neurons with the *Tmem63^{KO}* allele restored the attraction behavior to 70% RH. $n = 10, 9, 11, 12, 11$ assays. Two-tailed unpaired *t* test. For (**c**, **d**), data points are mean values and shaded area represents ±SEM. For (**f**, **g**), data are mean ± SEM. Source data are provided as a Source Data file.

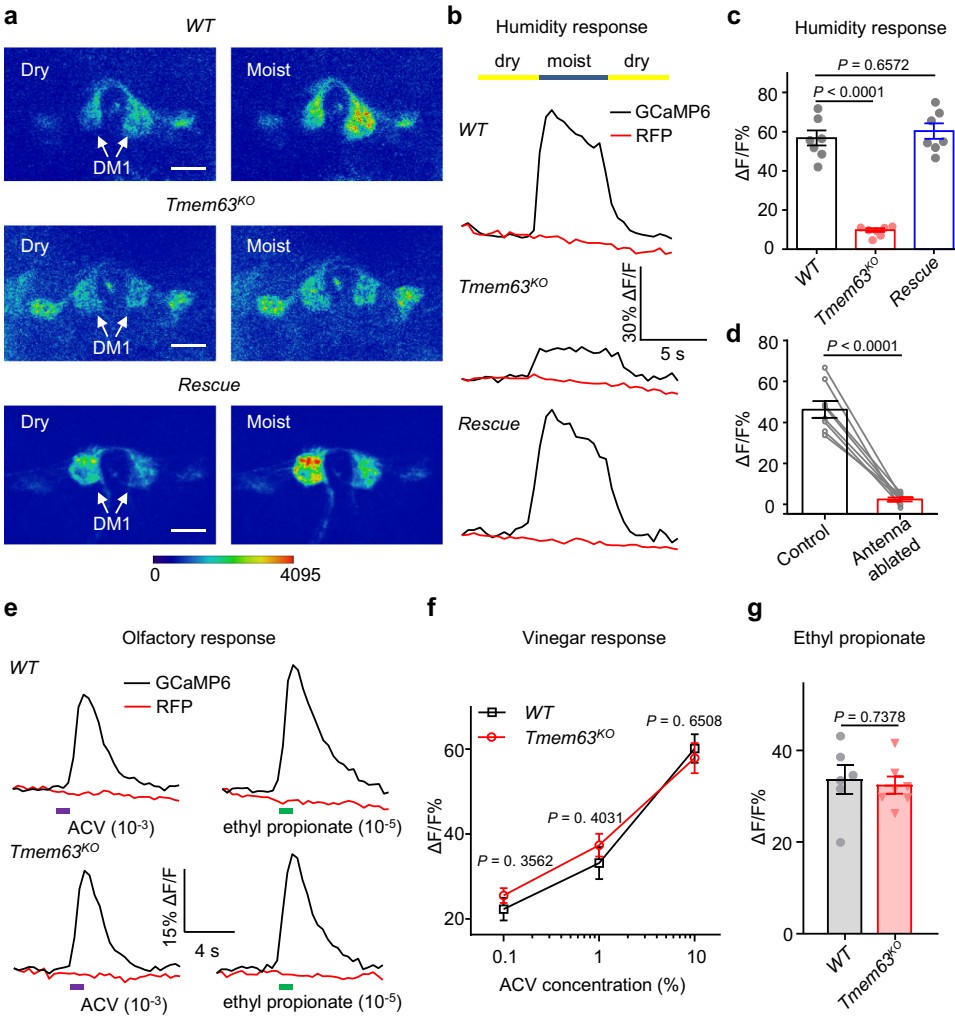

**Fig. 5 Tmem63-dependent calcium response to humidity changes in Or42b neurons. a** Pseudocolored images of $Ca^{2+}$ dynamics in response to humidity changes for the *wild type* (*Nan-Gal4/+; UAS-GCaMP6m,UAS-tdTomato/+*), *Tmem63KO* (*Nan-Gal4,Tmem63KO/Tmem63KO; UAS-GCaMP6m,UAS-tdTomato/+*) and *Rescue* (*Nan-Gal4,Tmem63KO; UAS-GCaMP6m,UAS-tdTomato/UAS-DmTMEM63*) animals. White arrows denote the DM1 glomerulus. Scale bar, 20 μm. Each result was repeated over three times. **b** Representative traces of the $Ca^{2+}$ response of the DM1 glomerulus to humidity changes in *wild type*, *Tmem63KO* mutant and *Rescue* flies. **c** Statistical analysis of the $Ca^{2+}$ response in the Or42b neurons of *wild type*, *Tmem63KO* and *Rescue* flies to a switch from dry air to moist air. $n = 7$ flies for each group. One-way ANOVA followed by Dunnett's test. **d** Humidity responses in the DM1 glomerulus prior to and after surgical resection of the bilateral antennal nerve. $n = 8$ flies. Two-tailed paired $t$ test. **e** Representative traces of the $Ca^{2+}$ response of the Or42b neurons in *wild type* (upper panel) and *Tmem63KO* (lower panel) flies to $10^{-3}$ dilution of vinegar (magenta bar) and $10^{-5}$ dilution of ethyl propionate (green bar). **f** Statistical analysis of the $Ca^{2+}$ response in the Or42b neurons from *wild type* and *Tmem63KO* flies to apple cider vinegar in different concentrations. $n = 7, 6$ flies. **g** Quantification of peak $\Delta F/F_O$ response to ethyl propionate. $n = 6, 7$ flies. **f, g** Two-tailed unpaired $t$ test. Data are mean ± SEM. Source data are provided as a Source Data file.

Moreover, distinct *Drosophila* species show diverse humidity preferences and therefore have different geographical distributions, for example, experiments showed that rainforest flies prefer a high level of humidity while desert-dwelling flies prefer 20% RH[6,12]. Apart from insects, other poikilotherms also display similar humidity-related geographical distributions[1,2,12]. Our findings may provide a clue to study how distinct animal species evolve to select their native habitats.

Our data show that TMEM63 mediates hygrosensation in Or42b neurons, a neuron group for food odor detection in *Drosophila*, suggesting an early integration of hygrosensory and olfactory inputs. This is not surprising since insects rely on multisensory integration for locating vital resources[49–51]. Disease-transmitting mosquitoes also exploit multimodal cues, such as $CO_2$, odors, body heat and moisture, for finding and selecting potential hosts[9,52]. Molecular receptors for chemicals and

temperature have been unraveled in mosquitoes[51,53–55], but the receptors that incorporate humidity into multimodal sensory inputs have remained elusive. It will be interesting to test if *Tmem63* functions as a humidity sensor in mosquitoes and guide host attraction by detecting the elevated humidity levels at the proximity of the host, which might provide insights into the molecular underpinnings of host approach behavior in disease vectors.

The TMEM63 proteins have been found to be evolutionarily conserved from *Drosophila* to mammals[38–40], but research on their physiological function is still in its infancy[46–48]. We showed that human TMEM63B can substitute for *Drosophila* TMEM63 to mediate moisture attraction behavior, an evolutionary solution to satisfy the internal needs for water. Further studies are necessary to determine whether mammalian *Tmem63* genes function in behaviors related to osmotic regulation or water homeostasis.

## Methods

**Fly stocks**. Or42b-Gal4, Or42b-Gal80, Nan-Gal4, Ir31a-Gal4, Ir41a-Gal4, Ir40a-Gal4, Ir75a-Gal4, Ir8a-Gal4, R11F02-Gal4, $y^1,w^*$,UAS-mCD8-RFP,LexAop2-mCD8-GFP, Orco[2], Or42b[EY14886], piezo[KO], nompC[1], IR68a[MB05565], IR25a[2], Df(2R)7094, LexAop-GCaMP6m, LexAop-Gal80, UAS-Kir2.1, Orco-RFP, UAS-RedStinger, UAS-CD4-tdGFP, UAS-GCaMP6m and UAS-tdTomato were obtained from the Bloomington Stock Center. UAS-Tmem63-RNAi and UAS-Dicer2 flies were obtained from the Vienna Drosophila Resource Center. UAS-ReaperHid was a gift from Hermann Steller lab. Ir68a-Gal4 was a gift from Paul Garrity lab. iav[1], tmc[GAL4], nompC[f00914] and UAS-pzl-RNAi were gifts from Wei Zhang lab. UAS-ppk-RNAi and UAS-nan-RNAi were from Tsinghua Fly Center. ppk28[Δ] was from the Core Facility of Drosophila Resource and Technology in Shanghai Institutes of Biochemistry and Cell Biology. The flies were raised on standard medium at 25 °C and 60% humidity under a 12 h/ 12 h light-dark cycle.

**S2 cell culture and transient transfection**. Drosophila S2 cells were cultured in Schneider's medium supplemented with 10% FBS at 25 °C. Cells were plated into 35 mm petri dishes before transfection. TransIT-Insect (Mirus) was used to transfect cells according to the product instructions. All constructs with the pUAST backbone were cotransfected with pActin-Gal4.

**Generation of Tmem63[KO] mutants and knock-in reporter lines**. We used a previously described targeting strategy[41] to obtain the Tmem63[KO] mutant allele with an attp site introduced to the first intron, then reporter constructs were integrated into the attP site through phiC31 mediated gene integration to generate the corresponding reporter lines.

The Tmem63[KO] mutant lines were generated through homologous recombination in Drosophila embryos via the CRISPR/Cas9 system. The 5′ and 3′ homologous arms of Tmem63 were cloned from nos-Cas9 flies by PCR amplification.

Primers for 5′ homology arm amplification:
5′-GATGCCAGAACAAATACATCGAGA-3′ and 5′-GATTGTTATCCGCT TTCATAGCAC-3′

Primers for 3′ homology arm amplification:
5′-ACTTTACACCGGGCACCTACC-3′ and 5′-TTGCCTTGCTGTTATC CTCCTG-3′.

To generate the knock out targeting vector, the 5′ arm and 3′ arm flanking an attP-3P3-RFP-loxP cassette were cloned into pBSK+ vectors using a MultiS One Step Cloning Kit (Vazyme). The sgRNA sequences were as follows:
sgRNA1: 5′-AAATAAGTGAATGCGACGAA-3′;
sgRNA2: 5′-GTTCTGGGCCTCCTGCACAG-3′.

The guide RNAs were expressed under U6b promoter control. A mixture of the targeting vector and two sgRNAs was injected into Nos-Cas9 embryos. F1 flies with RFP-positive eyes were selected as mutant candidates and verified by genotyping. The mutant was confirmed by RT-PCR, with the Actin5C gene as a positive control. The primer pairs were as follows:
Tmem63:
forward-primer, AGGGATACTCTCATGGTTAAACCAG;
reverse-primer, TCCGAGGGAAGGCAGAATCATG.
Actin5C:
forward-primer, CTGGGACGATATGGAGAAGATC;
reverse-primer, CAGCTCGTAGGACTTCTCCAAC.

To generate the knock-in marker line, the replaced sequence and the rest genomic region of Tmem63 before the stop codon was amplified from w[1118] flies and cloned into the pBSK-attB-EGFP-T2A-Gal4 and pBSK-attB-V5-T2A-LexA vectors[41] using the ClonExpress II One Step Cloning Kit (Vazyme). These vectors were then separately injected into embryos from nos-phiC31 females crossed with Tmem63[KO] males. Knock-in Tmem63[LexA] and Tmem63[EGFP-Gal4] lines were obtained from F1 flies with red eyes and verified by PCR. Finally, the knock-in lines were crossed to hs-Cre flies to remove screening markers in the genome.

**Generation of transgenic flies**. The full length cDNA of Drosophila Tmem63 (RE44586) was purchased from the Drosophila Genomics Resource Center, and the full length cDNA of human Tmem63a (NM_014698.3), Tmem63b (NM_001318792.1) and Tmem63c (NM_020431.4) were synthesized (Genewiz). The coding regions of these genes were obtained by PCR and subcloned into the pUAST vector using the ClonExpress II One Step Cloning Kit (Vazyme). UAS-DmTMEM63, UAS-HsTMEM63A, UAS-HsTMEM63B and UAS-HsTMEM63C transgenic flies were generated by conventional P-element-mediated germ-line transformation.

The 5 kb Or42b gene promoter was PCR amplified from the genomic DNA of Or42b-Gal4 flies. The CD8-RFP coding region following the Or42b promoter were assembled to pBSK-attb vector via MultiS One Step Cloning Kit (Vazyme). Or42b-CD8-RFP flies were generated by phiC31 mediated gene integration to the attP2 site on the third chromosome.

**Humidity-guided attraction behavior assay**. The humidity-guided attraction behavior assay was modified from the hygrotaxis assay[18,19]. In total, 20–40 male flies aged between 7 and 10 days that had been sorted into groups 2 days before testing were placed in a 60 mm Petri dish that was covered with 250 mesh nylon net. Petri dishes with flies were placed in a sealed chamber with desiccant and desiccated for 6 h prior to the test; no food was provided during desiccation. The dish with flies under desiccation stress was placed above a 24-well plate with 5 holes immediately beneath the dish. Different humidity levels were created with distilled water or saturated salt solutions[23]. The moist hole on one side provided high humidity, while the hole on the opposite side provided low humidity, which results in a humidity gradient between the two specific humidity levels. The room humidity was kept at 50% to 60% RH with household humidifier or dehumidifier during testing. The room temperature was set at 25 °C. Assays were performed during 17:00 and 22:00. The water-deprived flies were allowed to walk for 2 min, and the petri dishes were videotaped at 30 frames per second. The attraction index was then calculated for every 10 s time point as follows (the attraction index for 0 s was defined as 0.12, calculated from a random distribution: the area of the 20-mm-diameter moist well divided by the area of the 60-mm-diameter arena):

Attraction index = Number of flies above moist hole / Total number of flies

The locomotor activity of the thirsty flies was assayed by using a previously established video-assisted tracking method[56]. Two male flies were introduced to a 60 mm petri dish for 6 h of desiccation and then videotaped at 30 frames per second for 2 min under the humidity conditions in which the flies were raised.

For quantification of the attraction index for distinct groups, an average index of the attraction indexes from the plateau stage was used. For spatial distribution analysis and locomotion analysis, flies were first tracked using Flytracker[57] (http://www.vision.caltech.edu/Tools/FlyTracker) to analyze the positions and movement trajectories of the flies during the assay. Then, the data were processed and plotted using custom programs and scripts in MATLAB based on generic codes from Plotly (https://plotly.com/matlab/2D-Histogram). The whole arena was divided into 20 ×20 pixels, and the total time that a fly spent on each pixel during 80–90 s after the onset of the assay was calculated as the cumulative time. Pixels with a cumulative time above 10 s suggest more than one fly staying in the same pixel.

**In vivo calcium imaging**. Antennal lobe calcium imaging was performed at room temperature (25 °C) as described previously[13,42]. The room humidity was also kept at 50% to 60% RH with household humidifier or dehumidifier during experiments. Briefly, 3- to 5-day-old flies were mounted in a custom made stage using two component silica gel. The antennae were pushed toward with a thin sheet of plastic on top of the head. Then a small window was made in both the plastic sheet and the head capsule to allow access to the antennal lobes. A drop of Adult Hemolymph-Like Saline (containing in mM: 2 CaCl$_2$, 5 KCl, 5 HEPES, 8.2 MgCl$_2$, 108 NaCl, 4 NaHCO$_3$, 1 NaH$_2$PO$_4$, 10 sucrose, 4 trehalose, pH 7.5) was then added to cover the imaging window. For the humidity stimulation, airstreams from a 50 mL syringe were passed through either a gas washing bottle with desiccant or a gas washing bottle containing a water soaked sponge, generating a 24.66 ± 0.62% RH dry airstream or a 66.67 ± 0.19% RH humid airstream, respectively[13]. Humidity changes were achieved by alternating the two syringes producing the dry airstream or humid airstream. Odors were obtained from filter paper with the corresponding odor source in a gas bottle as previously described[32,42]. The required dilution of ACV in water or ethyl propionate in mineral oil was used as the odor stimulus. The airflow rate was ~500 mL/min. We used flies carrying Nan-Gal4 (ref. [36]), an alternative driver for the Or42b neurons, to express GCaMP6 and RFP for calcium imaging. Images were acquired with FV10-ASW 4.2 software from an Olympus FV1200 confocal microscope equipped with a 10x water-immersion objective. GCaMP and red fluorescent proteins (as references) were excited by a 473-nm and a 559-nm laser, respectively. The average GCaMP signals from the first 3 s before the introduction of the odorant stimulus or a humidity change were taken as $F_0$, and $\Delta F/F_0$ was calculated for each data point.

**Measurement of sensilla shape change**. One day old adult flies carrying Or42b-Gal4 and UAS-CD4-tdGFP were used to perform confocal microscopy imaging. Acutely isolated antennae were mounted on a slide using double-sided tape (Scotch 3 M). A coverslip was placed onto the slide using spacers so that a thin tube (~0.2 mm diameter) can be inserted for the humidity stimulation. Humidity stimulation was applied as described in calcium imaging experiments. The room temperature was set at 25 °C and room humidity was set to ~50% RH. Images were obtained at the region of the GFP-positive sensillum with FV10-ASW 4.2 software on an Olympus FV1200 confocal microscope equipped with a 60x water-immersion lens at ~2 Hz.

The curvature analysis is conducted using the Kappa plugin of Fiji[58] according to the documentation on Github (https://github.com/brouhardlab/Kappa). We first aligned the image stack using the StackReg plugin of Fiji, only the image stacks in which the sensilla base displayed no movement were opened in Kappa. The time point in which the dry air was converted to moist air was defined as 0 s. We chose the images at time points −5 s, 2.5 s and 10 s for the next analysis. In Kappa plugin, five points that track the midline of the sensilla were first manually defined to make an initial curve. We then adjusted parameters such as the color channel, the brightness threshold, and the distance from the curve to make all GFP signals or bright pixels within the sensilla contour selected. Next, we fitted the initial curve to the chosen pixels by the least square algorithm in Kappa. After setting the scale

factor (μm/pixel) of the images, we exported the average radius of curvature of the fitted curve for statistical analysis.

**Immunostaining**. Whole-mount antennal staining was performed as previously described[12]. Male 2- to 5-day-old flies were dissected in Adult Hemolymph-Like Saline, and the 3rd antennal segments were fixed in 4% PFA at room temperature for 5 min. Brains of 5- to 7-day-old flies were dissected in 1x PBS and fixed in 4% PFA at room temperature for 20 min. After three washes, the samples were blocked in block buffer (5% normal goat serum with 0.3% Triton in 1x PBS) for 30 min at room temperature. The samples were then incubated with primary antibody at 4 °C overnight. On the second day, tissues were washed three times and incubated in secondary antibodies at room temperature for 2 h. All samples were mounted in Rapiclear 1.47 (SunJin Lab) for confocal microscopy (Olympus FV1200). The following antibodies were used: rabbit anti-GFP (1:200, Proteintech 50430-2-AP), mouse anti-nc82 (1:20, DSHB), donkey anti-rabbit Alexa488 (1:500, Jackson ImmunoResearch), donkey anti-mouse Alexa647 (1:500, Jackson ImmunoResearch).

For *Drosophila* S2 cell staining, we used a PCR-based approach to introduce the in-frame fused myc-tag (EQKLISEEDL) behind the F21 or G724 site of *Dm*TMEM63. The pUAST-21myc-Tmem63-mCherry or pUAST-724myc-Tmem63-mCherry construct was cotransfected with pActin-Gal4. 24-36 h after transfection, cells were plated onto ConA-coated coverslips for staining.

For nonpermeabilized staining, the primary antibody (mouse anti-myc-tag, Cell Signaling 2276) was diluted 1:200 in Schneider's *Drosophila* Medium and incubated with transfected cells for 30 min at 25 °C. After fixation with 4% PFA for 30 min at 4 °C, the cells were blocked for 30 min at room temperature and then incubated with the secondary antibody (goat anti-mouse Alexa488, 1:500, Jackson ImmunoResearch) for 30 min.

For permeabilized staining, cells were fixed and incubated with PBST (PBS + 0.1% Triton) for 10 min. Then the cells were blocked and stained with the primary and secondary antibodies.

**S2 cell electrophysiological recordings**. S2 cells were transfected with *Dm*TMEM63-GFP or GFP empty vectors and incubated for 24-36 h before recording. Outside-out patch recordings of S2 cells were carried out at 25 °C under an Olympus BX51WI microscope equipped with a 40x water-immersion lens as described before[59]. Recordings were performed using borosilicate glass pipettes with resistances of around 10 MΩ. The pipette solution contained 140 mM potassium gluconic acid and 10 mM HEPES. The bath solution contained 140 mM NaMES (sodium methanesulfonate) and 10 mM HEPES. All solutions were adjusted to 320 mOsm and pH 7.2. Negative pressure was applied to the excised membrane patches using a HSPC device (ALA-scientific). The sample rate was 20 kHz and filtered at 1 kHz (low-pass). A multiclamp 200B amplifier, DIGIDATA 1550 A and Clampex 10.5 software (Molecular Devices) were used to acquire and process the data.

**Humidity measurements**. Humidity was measured using an custom 2.5 × 2.5 mm probe equipped with SHT31 sensor and recorded with the SHT31 Smart Gadget (Sensirion). To monitor the humidity gradients formed in the humidity-guided attraction behavior assay, the arena was divided into 10 × 10 measurement points to allow the insertion of humidity sensor for monitoring the humidity distribution over a 120 s time period.

**Statistics and reproducibility**. Experimental animals and genetic controls were tested at the same condition, and data were collected from at least three independent experiments. Statistical analysis were carried out in Prism 7 (GraphPad). Statistical methods used include two-tailed *t* test, one-way ANOVA followed by Holm-Sidak or Dunnett's post hoc test, or the two-way ANOVA with Sidak's multiple comparison test. Post hoc power analyses were performed in PASS 15 (https://www.ncss.com/software/pass/) to ensure that statistical power >0.8 for all the significant differences. All data in bar and line graphs are presented as means ± SEM and the exact *P* values are displayed in the Figures.

**Reporting summary**. Further information on research design is available in the Nature Research Reporting Summary linked to this article.

## Data availability

All data generated in this study are available within the article and its Supplementary Information files. Any additional data and information are available upon request to the corresponding author, Dr. Zhiqiang Yan. Source data are provided with this paper.

## Code availability

Analyses were performed with MATLAB programs based on generic codes from Plotly (https://plotly.com/matlab/2D-Histogram). The code for Flytracker is available from Github (https://github.com/kristinbranson/FlyTracker).

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

## Acknowledgements

We thank Dr. Davide Filingeri, Dr. Minmin Luo and Dr. Richard Benton for their comments on the experimental design and the manuscript. We thank Yuh-Nung Jan at UCSF, Paul Garrity at Brandeis University and Wei Zhang at Tsinghua University for the fly lines, Bowen Deng and Yi Rao at Peking University for technical help with generation of CRISPR mutants. We also thank the Core Facility of *Drosophila* Resource and Technology (Shanghai Institutes of Biological Sciences, Chinese Academy of Sciences) for the fly microinjections. The research was supported by funds from China Brain Project (2021ZD0203304), Shenzhen Science and Technology Program (RCJC20210609104631084), the National Key R&D Program of China Project (2021YFA1101302, 2017YFA0103900, 2016YFA0502800), the National Natural Science Foundation of China (31571083, 31970931), the Program for Professor of Special Appointment (Eastern Scholar of Shanghai, TP2014008), the Shanghai Municipal Science and Technology Major Project (No.2018SHZDZX01), ZJLab and Shanghai Center for Brain Science and Brain-Inspired Technology, and the Shanghai Rising-Star Program (14QA1400800).

## Author contributions

Z.Y. conceived and designed the project. Z.Y. supervised the project throughout. S.L. designed and performed most experiments. B.L. performed electrophysiological recordings and the calcium imaging experiments. L.G. and J. W. helped with some experiments. S.L.,B.L. and L.G. performed data analysis. S.L., B.L. and Z.Y. wrote the manuscript.

## Competing interests

The authors declare no competing interests.
