## [Peer Review File · Nature Communications]

Humidity response in *Drosophila* olfactory sensory neurons requires the mechanosensitive channel TMEM63Reviewers' comments:

Reviewer #1 (Remarks to the Author):

The submitted manuscript by Li et al. concerns the identification of TMEM63 in the sensation of humidity (Hygrosensation) in olfactory neurons in the fruit fly *Drosophila melanogaster*. With a screen of desiccated flies they identify OR42b neurons as partially necessary for attraction to a humid area. They go on to show potential humidity-induced deflection of the sensory hairs that include OR42b sensory cilia. Next, they perform a new screen of flies with mutations in genes encoding mechanically gated proteins and identify TMEM63 mutants as deficient in their behavior assay. The phenotype can be rescued by reintroducing TMEM63 (Both *D. melanogaster* and *Homo sapiens* TMEM63B) exclusively into OR42b neurons suggesting TMEM63 functions exclusively in these cells for this behavior. Next, they perform calcium imaging and demonstrate that Or42b neurons show an increase in calcium when a dry air stimulus is switched to a moist and in TMEM63 mutants this response is reduced. Finally, they knockdown TMEM63 in the mosquito *Anopheles stephensi* and show that these mosquitos show a slightly lower/delayed attraction to a human hand.

The Or42b neurons are well studied and are considered as 'fermentation detectors' responsible for attraction toward vinegar smells in decomposing fruit. In the numerous studies on the response profile a water stimulus (increased humidity similar to the one used in this paper) has often been performed (e.g. PMID: 23457557, PMID: 20530374 more references can be found here, search for Or42b <http://neuro.uni-konstanz.de/DoOR/default.html>). None of these studies detected a response to a water stimulus. One preprint (<https://www.biorxiv.org/content/10.1101/2020.08.09.242784v2>) even report that Or42b neurons respond to methanol and ethanol, two highly hygroscopic chemicals that would produce a dry stimulus. Therefore, the manuscript of Li et al present data that goes against the conclusion of a large body of literature. To go against dogma this way requires exceptional attention to detail, something this manuscript lacks. So after reading this I am not convinced that the Or42b neurons are responsive to humidity. TMEM63 is surely expressed in Or42b neurons, it's a mechanically-gated receptor, but it's unlikely to transduce humidity changes. I recommend the authors reconsider their hypothesis on the function of TMEM63.

Specific points:

- Relative humidity (RH) is used to refer to humidity levels throughout. However, RH is not an informative value if it is not also accompanied by the temperature value. Please add the temperature each experiment was performed at.
- Figure 1:
 - o OR42b-neurons are identified in a Gal4 screen, but only the data for Or42b-Gal4 is presented. What other lines were included in this screen? It is of general interest to the reader to see how other Gal4-line performed in the assay. Especially since the phenotype from Or42b-Gal4>ReaperHid/Kir only produce a mild reduction in attraction index. Is the rest of the phenotype accounted for by Ir25a-Gal4? Orco-Gal4? How does a humidity-blind fly like IR25a2 or IR93aMI05555 perform in this novel assay? It makes a difference for the reliability of the results if IR25a2 or IR93aMI05555 are humidity-blind (or not) also in this assay.
 - o The flies used in the assay have been starved and desiccated. Since Or42b neurons are associated with food search it would be of interest to see how sated behave in the arena? Or

starved but not dessicated (Grown on pure agar a couple hours)? How does that influence the phenotype of Or42b-Gal4>ReaperHid/Kir?

o Figure 1E error bars are not visible

• Figure 2:

o The presented mechanism for humidity-induced structural changes in olfactory sensilla strikes me as unlikely. ab1 sensilla are exposed on the surface of the antenna and a gust of wind would deflect the sensillum to a similar degree as the suggested humidity-induced changes. How would the fly differentiate between wind and humidity? Hygrosensilla in *D. melanogaster* and in other insects are located in positions where they are shielded from air (eg in the sacculus) for this reason.

o No statement about humidity levels are presented in this figure. At what RH and temperature where the experiments performed and how much was the humidity changed to induce the deflection?

o Only dry -> moist stimulus was used. Adding moist -> dry would be easy to do and would give valuable data. It would be expected to return the change in deflection and curvature.

o Do other sensory hairs, especially basiconic, change shape in a similar way as ab1? As far as I know all sensilla are composed of similar material (chitin) that would interact with water vapour in similar ways as ab1. Does this mean all basiconics are responsive to humidity?

o The example images used in this figure look very blurry. How can you accurately measure such fine changes in deflection (only three degrees) and curvature from this material? I would recommend trying calcofluor or UAS-ChtVis-Tomato (PMID: 26395478) for better contrast. But for best contrast and analysis you would probably need to study these effects using electron microscopy.

o More details about how the measurement of deflection and curvature should be included in the methods section. How was deflection angle calculated?

• Figure 3:

o TMEM63 expression is showed for a thin section of the antennal lobe. Which other glomeruli are TMEM63 expressed in? It looks in the image as the V-glomerulus is labelled. Assuming a humidity-induced change in shape of ab1 is trasduced by TMEM63 expressed in ab1A-neurons, a similar response would be seen in ab1C-neurons if they also express TMEM63. Are ab1C also responsive to humidity?

o Is TMEM63 expressed in any of the known humidity-responsive structures the sacculus/VP-glomeruli?

o No error bars for control in 3f or 3g.

• Figure 4:

o How was humidity stimulus delivered for the calcium imaging experiments? Since ab1 sensilla seems to be sensitive for deflection it would be important there is no change in flow rate when stimulus is changed. As the methods section is written it is not clear how the flow was generated, just that it was created by a syringe. What pressed the syringe? A flow rate of 500 ml/min is given, how, where and when was this measured?

o As I wrote above, the ab1A neurons have been tested with a water stimulus before with no response and with a response to a putative dry stimulus (methanol/ethanol). How can these discrepancies be explained? The calcium imaging data in this paper shows that the Or42b neurons even respond more strongly to moisture than to apple cider vinegar. Does this mean humidity is the primary stimulus these neurons are responding to? What happens if you present apple cider vinegar at different humidities? Does it modulate the response on the Or42b neurons?

- o When activated by vinegar smells, Or42b-neurons signal approach behavior (PMID: 19396157) and here Or42b neurons are shown to be activated by humidity even more strongly than vinegar. Therefore, a humid stimulus would generate a stronger approach behavior than apple cider vinegar. A suggested experiment would be the humidity behavior arena from Figure 1a with moisture in one test well and apple cider vinegar in the other test well. Will the flies go for moisture (as this data indicates) or apple cider vinegar. The experiments can be performed in Ir25a2 flies where the sacculus neurons are not functioning, so they won't impact the outcome.
- o Error bars are missing from the traces in 4b
- Figure 5:
 - o The humidity measurement in 5b does not seem to represent experimental conditions. In the experiment five breaths are given before the hand is presented. How do these breaths change the humidity? I assume humidity (and temperature) would be increased? A better way of performing these experiments would be to have a constant stream of CO₂ to activate search behavior instead of giving breaths.
 - o The effect seen in 5d is very small. Based on this, can you really state that host approach in mosquitos "requires Tmem63"? It seems like an overstatement to me. The mosquito part of this manuscript deserves its own separate study to be able to say anything about mosquito hygrosensation.

Reviewer #2 (Remarks to the Author):

The study by Li et al. reports on the identification of a subpopulation of olfactory sensory neurons (Or42b) as mediators of hygrosensation using *Drosophila* as a model system. The study addresses an open question on how hygrosensors transduce humidity signals into electrical signals that drive behavioral responses by implicating mechanical deformations of the cuticular sensilla housing Or42b neurons via the action of the mechanosensitive ion channel TMEM63 within these neurons. The authors document evolutionary conservation of TMEM63 orthologues via cross-species rescue experiments with human TMEM63B and present evidence for a role of mosquito TMEM63 in host-seeking via humidity cues.

A major strength of this manuscript is the insight into the mechanisms by which humidity signals are transformed into electrical activity to drive stimulus-relevant behavior by providing evidence in support of a mechanical hygrometer model via the action of TMEM63. Overall, the work is largely compelling, carefully performed and should prove of interest to others in the field. The development of a novel behavioral assay should also facilitate future studies to glean additional insights in the molecular control of this process coupled with potential insights into putative targets for disease vectors that utilize humidity cues.

The statistical analyses and descriptions of the methodologies and approach are clear and supported.

Below I summarize my primary critiques/comments on the manuscript that the authors should address:

Critiques/Comments:

Page 3, line 13: The authors state that wild type flies showed robust moisture attraction within 90 seconds. This is certainly true, though the data in Fig. 1C show that attraction response appears to peak much earlier at 50 sec.

Page 4, line 12-14: What is the nature of the Or42b[EY] allele with respect to disruption of this gene? Can the authors exclude a role of olfactory receptors entirely in hygrosensation as the sentence reads vs. editing the sentence to reflect that the data suggest that Or42b is not involved in hygrosensation pending clarification on the nature of this EY allele.

Page 4, Lines 26-28 and Supplementary Figure S3: The authors show that iav mutants perform normally in their assay. How do the authors square these results with previous studies showing that iav mutation affects hygrosensation? (Liu L et al, 2007. doi: 10.1038/nature06223). This should at least be addressed in the text.

Page 6, Line 26: The authors should specify the odorants in the results text. Also, the statement that it is “dispensable for odor detection” is too broad. If the authors think the use of apple cider vinegar and ethyl propionate support a broad interpretation, they should specify why, otherwise the statement should be modified to reflect that this is only true for the odorants specifically tested.

Page 23, line 10: I am not entirely clear on what the authors mean by “results from one of three independent experiments are shown” given that the data in Fig. 5a display SEM bars?

Fig. 1: What is the efficiency of the Or42b ablation using ReaperHid? Can the authors show data demonstrating this in combination of Or42b labeled neurons?

Fig. 2: What is the time scale for the change in curvature observed upon exposure from dry-to-moist humidity? How long does the mechanical deformation last? In Fig. 2c, how do the authors explain the difference in baseline radius of curvature for dry conditions given that the significant increase in curvature observed with dry-to-moist appears due to the fact that the baseline dry average curvature is lower?

Fig. 4: For the GCaMP6 experiments performed in Fig. 4, why do the authors switch from using Or42b-GAL4 to using Nan-GAL4? Can the authors provide data to show the overlap between these two drivers? In the Fig. 4 legend, the authors refer to the calcium responses in Or42b neurons, but do not sufficiently establish that Nan-GAL4 labels the same population of neurons. Also, in the Fig. 4 legend, the genotypes do not include the RFP normalization transgene.

Supplementary Figure S3: Why did the authors choose different molecule subsets for the mutant and RNAi approaches?

Reviewer #3 (Remarks to the Author):

General comments

In this study, Li et al. reports structural and molecular evidence for mechanosensitive molecules to act as hygrometers and mediate hygrosensation in *Drosophila*. The Authors suggest that this evidence supports the mechanical hygrometer model, in which humidity changes can be transformed into mechanical cues, subsequently used by *Drosophila* to locate environments with preferred humidity levels. A secondary question involved investigation of whether a mechanosensitive molecular homologous in the mosquito vector is also involved in this insect's host seeking behaviour.

The Authors approaches a fundamental question in environmental sensory transduction which is of broad interest, yet still relatively little understood (i.e. when compared to thermosensation). The study involves sophisticated genetic, molecular, structural, and electrophysiological methods. Furthermore, some of the results are novel, and they certainly add to the understanding of the molecular architecture of hygrosensation in invertebrates. However, I have several concerns regarding the validity of the modified hygrotactic assay reported by Li et al., and its suitability (in current form) for providing the conclusive evidence the Authors ultimately rely upon for their claims.

While the genetic, molecular, structural, and electrophysiological data are important mechanistically, the behavioural data are the primary outcomes here; hence, it is my opinion that the validity and reliability of the Authors' new assay is of paramount importance to ensure that the mechanistic evidence can be confidently relied upon.

My first, main question assessing this study was to determine whether flies showed a sufficiently robust and (most importantly) a clear humidity-dependent-behaviour. Unfortunately, looking at some of the data reported (e.g. attraction indices under normal conditions no greater than 60% and under disrupted conditions no smaller than ~30%) I feel the latter may not be the case.

Secondly, there are important information about the assay which are missing, i.e. what were the temperature conditions within this and where temperature gradients present? I feel this information is particularly important to better untangle the multisensory nature of this hygrotactic behaviour, that is, what is the complementary role of thermosensory cues in hygrosensing, as previously reported (see also next comment).

Thirdly, and perhaps most importantly, it seems to me that the evidence reported indicates that while important, TMEM63 is not essential for hygrosensing, i.e. its absence does not completely abolish hygrosensing. Indeed, under *Tmem63*KO conditions, the attraction index went from ~60 to ~30% (Fig. 3), indicating that this molecular disruption only halved the likelihood of attraction (i.e. 30% of flies still gathered in the 70% RH, which is roughly three times as much as the 12% random distribution). Accordingly, I would have expected the authors to perform concurrent experiments to determine the complementary role of thermosensory molecules in the remaining hygrosensing function in their flies, in a way similar to the approach taken by Russell et al. with the *C.elegans* (see <https://www.ncbi.nlm.nih.gov/pmc/articles/PMC4050571/>). As the Authors also report when discussing the mosquito data, it may be likely that flies also integrate multisensory cues in hygrosensing (in a way similar to what humans do, see

<https://pubmed.ncbi.nlm.nih.gov/24944222/>); hence, concurrent thermo- and mechano-sensory experiments could have shed a better light on the role of multiple hygrosensing molecules in this fly's behaviour.

All in all, I believe that, while certainly providing interesting data, the main conclusions of the current manuscript rely on experimental evidence that is not as strong as one would hope for. Accordingly, I would invite the Authors to consider these general comments, along with my list of detailed queries reported below.

Detailed comments

1. Please introduce the model organism used in the title of the manuscript (i.e. "...mediate hygrosensation in *Drosophila*").
2. Please provide more information about the validity and reproducibility of this hygrosensing assay – what was the temperature gradient within the assay? Why did you opt for such a short duration assay (e.g. when compared to Enjin et al. where 4h were allowed, which resulted in attraction index of ~1)?
3. Why only 60% of flies were attracted to higher humidity? This is just above chance level (50%), and it opens to the question of why ~40% of flies under desiccation are not attracted by 70% humidity?
4. What is the independent role of desiccation stress on fly's behaviour (independent of humidity-seeking behaviour)? Would it not be more appropriate to have a type of fly with a clear preference for that humidity level, and tests this with/without antenna to determine the true stress-free role of those antennae? After all, Enjin et al. 2016 showed that wildtype flies orient naturally (e.g. to 70%RH) based on their habitat of choice.
5. This assay gives only 60% attraction (which goes down to 15% when removing the antenna). However, when silencing odour neurons, attraction goes down to 36% (a drop of only 24%). It seems clear to me that there is something missing in the assay to drive full moisture attraction and that those odour neurons contribute to only half of that attractive behaviour (is this evidence of a thermo-mechano integratory system for moisture detection, as showed in humans?). This is a critical point as it challenges the Authors' conclusion that: Our study provides structural and molecular evidence supporting the mechanical hygrometer model, in which humidity changes can be transformed into mechanical cues so that mechanosensitive molecules may act as hygroreceptors to mediate hygrosensation. I would therefore invite the Authors to re-consider this claim.
6. Why not incorporating an approach like that of Enjin et al.'s to look at the thermosensory component too? Russel et al. in PNAS (<https://www.ncbi.nlm.nih.gov/pmc/articles/PMC4050571/>) took this approach with the *C. elegans* and provided elegant evidence for a multisensory integration mechanism. I would have expected the Authors to develop an experimental model where thermal cues are silenced (to confirm previous data); then mechanical cues are silenced (the current novel approach); and finally both cues are removed (in a way similar to the antenna removal). This experimental

design would provide the most comprehensive and likely conclusive data on the fly's integratory mechanisms for hygrosensing. It is indeed important to note that, as also reported by Russell et al., moving across a humidity gradient can induce evaporative cooling which could provide thermosensory cues the animal may rely upon in their choice. This cannot be excluded in the current work, and may perhaps explain why half of KO flies retained humidity attraction?

7. Fig. 2c – the radius of curvature (as a summary of the deformation induced by a switch to moist air) has a different starting point (i.e. dry baseline) between dry-dry and dry-moist, yet identical end points. Hence, while the 2 conditions differ in their delta change, they have identical positioning under dry and moist conditions, which raises the question of whether humidity is actually being sensed via this mechanism? Why is the positioning different at baseline under the two dry conditions? I believe it is difficult to speculate that a hygroscopic movement has occurred in the dry-moist switch that was different to the dry-dry condition, (which the Authors use to justify the mechanical model). Furthermore, why are t-tests used for each comparison independently (dry.dry and dry.moist)? It would have been more appropriate to use a 2-way ANOVA with condition (i.e. dry-dry vs. dry-moist) and time (i.e. pre vs. post switch) as independent factors, to better ensure that baseline conditions would be equal prior to any change to sensilla cuticle positioning. It is my feeling that if this analysis were to be performed, no differences between conditions would be found. If that is the case, this is problematic, as it would further reduce the (already limited in my view) evidence that mechanical deformations in the cuticle have occurred under moist conditions and differently from dry conditions.

8. Can you confirm all mutant and rescued experiments were conducted with similarly desiccated flies?

9. I appreciate an attempt to link some of the fly work to the relevance for a malaria vector. However, the section on the malaria vector feels somewhat disconnected from the whole paper. The evidence reported is not as thorough as that for the fly, and the findings are in my view, even less robust. Specifically, the mosquito data show only a 20% reduction in host approach with the mechano-channel KO. Furthermore, I also question the validity of this assay as the % of mosquito approaching the host is at the chance level (50%) at the 20s mark (the point used in the Discussion to exemplify differences). After the 20s mark, the mutant mosquito still shows a host approach, which in fact reaches higher levels than what the control mosquito reached at the 20s mark. My question is therefore: is this mosquito's host approach really disrupted to a meaningful extent? I doubt it, and I would therefore invite the Authors to consider whether reporting this data may oversell their general findings.

10. There also very is limited information on the behavioural assay for the malaria vector, which hinders reproducibility (e.g. how many people where tested; how many repeats; etc.)

11. A primary point to consider in the Discussion is that the general evidence on mechano-integration for hygrosensing indicates a contribution of this mechanism at best, rather than strong evidence for the mechanical hygrometer model (as eventually claimed by the Authors).

12. A secondary comment on the Discussion is that the argument here should be about looking at how mechanosensory and thermosensory pathways contribute to hygrosensory strategies, rather than contrasting hygro- with thermosensory pathways (as the hygrosensory pathways

may well rely on both mechano AND thermo pathways).

13. I note no sample size a priori analysis was used. Why? And importantly, why not using an a posteriori analysis to determine the power achieved given the effects size of interest? In this regard, what is the minimum relative change in hygrosensing behaviour that is sufficient to determine a critical role of a specific molecule in that pathway?

14. Can the author provide accuracy values for their sensirion humidity sensors?

Davide Filingeri, PhD

Point by point response to the reviewers' comments:

Reviewers' comments:

Reviewer #1 (Remarks to the Author):

The submitted manuscript by Li et al. concerns the identification of TMEM63 in the sensation of humidity (Hygrosensation) in olfactory neurons in the fruit fly *Drosophila melanogaster*. With a screen of desiccated flies they identify OR42b neurons as partially necessary for attraction to a humid area. They go on to show potential humidity-induced deflection of the sensory hairs that include OR42b sensory cilia. Next, they perform a new screen of flies with mutations in genes encoding mechanically gated proteins and identify TMEM63 mutants as deficient in their behavior assay. The phenotype can be rescued by reintroducing TMEM63 (Both *D. melanogaster* and *Homo sapiens* TMEM63B) exclusively into OR42b neurons suggesting TMEM63 functions exclusively in these cells for this behavior. Next, they perform calcium imaging and demonstrate that Or42b neurons show an increase in calcium when a dry air stimulus is switched to a moist and in TMEM63 mutants this response is reduced. Finally, they knockdown TMEM63 in the mosquito *Anopheles stephensi* and show that these mosquitos show a slightly lower/delayed attraction to a human hand.

The Or42b neurons are well studied and are considered as 'fermentation detectors' responsible for attraction toward vinegar smells in decomposing fruit. In the numerous studies on the response profile a water stimulus (increased humidity similar to the one used in this paper) has often been performed (e.g. PMID: 23457557, PMID: 20530374 more references can be found here, search for Or42b <http://neuro.uni-konstanz.de/DoOR/default.html>). None of these studies detected a response to a water stimulus. One preprint (<https://www.biorxiv.org/content/10.1101/2020.08.09.242784v2>) even report that Or42b neurons respond to methanol and ethanol, two highly hygroscopic chemicals that would produce a dry stimulus. Therefore, the manuscript of Li et al present data that goes against the conclusion of a large body of literature. To go against dogma this way requires exceptional attention to detail, something this manuscript lacks. So after reading this I am not convinced that the Or42b neurons are

responsive to humidity. TMEM63 is surely expressed in Or42b neurons, it's a mechanically-gated receptor, but it's unlikely to transduce humidity changes. I recommend the authors reconsider their hypothesis on the function of TMEM63.

Response:

We appreciate the reviewer's summary of our work, though we have different opinion with the referee that our data has gone against the conclusion of a large body of literature. The humidity setting and humidity stimulations we used are quite different from prior studies (PMID: 23457557, PMID: 20530374), but it seems that reviewer #1 has ignored this important information. Or42b neurons respond to humidity changes rather than water stimulus. In our calcium imaging experiments, we used switches from $24.66 \pm 0.62\%$ RH dry airstream to $66.67 \pm 0.19\%$ RH humid airstream to generate humidity changes. It should be noted that $24.66 \pm 0.62\%$ RH condition was uneasy to achieve unless strictly controlled. We put these important points in methods but not the main text, so the reviewer might miss the information, which leads to the misunderstanding. For water stimulus, the pre-stimulus humidity was rarely strictly controlled and increased humidity might be too small to elicit neuronal responses. Using the SHT31 humidity sensor, we have further validated that a 2-s pulse of water stimulus gave rise to only subtle humidity change ($50.09 \pm 0.21\%$ to $52.05 \pm 0.35\%$ RH) when the ambient humidity is $\sim 50\%$ RH. However, applying a pre-stimulus dry airflow to the water stimulus caused a significant humidity change ($24.68 \pm 0.73\%$ to $41.66 \pm 0.40\%$ RH).

Moreover, although the nature of humidity is the water vapor content in the atmosphere, all humidity transduction models posit that humidity are indirectly detected by receptor neurons, which means water molecules are unlikely to activate humidity sensors by themselves (PMID: 29208217, PMID: 31657753). However, reviewer #1 might think that moisture sensing Or42b neurons should be responsive to water and established a connection between our data and previous studies involving water stimulus (PMID: 23457557, PMID: 20530374).

Importantly, when using the knock-in $Tmem63^{LexA}$ driver which labels multiple

antennal sensory neurons to conduct calcium imaging experiments, we observed that three olfactory neuron populations (including Or42b neurons) are responsive to a $24.66 \pm 0.62\%$ RH to $66.67 \pm 0.19\%$ RH humidity change, while two neuron groups (including the well established dry sensor Ir40a neurons) respond to the opposite humidity change. We have shown these data in the revised manuscript to support our conclusion (Fig. 3). The results are partially in accord with the records on DoOR (<http://neuro.uni-konstanz.de/DoOR/default.html>) with a query for odorant name “water”, both data reveal the DC4 glomerulus as a strong responding unit for moisture. Yet we noticed that some of these neuron groups have not been identified as humidity sensors previously, which might be due to the lack of sufficient humidity changes when using water stimulations.

Additionally, one preprint indeed reported that Or42b neurons respond to methanol and ethanol, two hygroscopic chemicals that would generate dryness in theory. If so, methanol and ethanol should activate the sacculus dry sensor Ir40a neurons. However, Extended Data Figure 1 in a previous report (PMID: 27337300) showed that both of the chemicals were unable to activate Ir40a neurons, suggesting that methanol and ethanol cannot produce dry stimulus. These two chemicals might act as an “odorant” stimulus but not a “dry” stimulus to activate Or42b neurons.

Consequently, we hope reviewer #1 would reconsider the overall evaluation of our work after reading the above responses. Below are our specific responses to the reviewer’s comments.

Specific points:

- Relative humidity (RH) is used to refer to humidity levels throughout. However, RH is not an informative value if it is not also accompanied by the temperature value. Please add the temperature each experiment was performed at.

Response:

Thanks for the suggestion. All behavioral and imaging experiments were performed at room temperature (25° C) and room humidity was controlled at 50-60% RH. We have added the relevant information in the method section.

- Figure 1:

o OR42b-neurons are identified in a Gal4 screen, but only the data for Or42b-Gal4 is presented. What other lines were included in this screen? It is of general interest to the reader to see how other Gal4-line performed in the assay. Especially since the phenotype from Or42b-Gal4>ReaperHid/Kir only produce a mild reduction in attraction index. Is the rest of the phenotype accounted for by Ir25a-Gal4? Orco-Gal4? How does a humidity-blind fly like IR25a2 or IR93aMI05555 perform in this novel assay? It makes a difference for the reliability of the results if IR25a2 or IR93aMI05555 are humidity-blind (or not) also in this assay.

Response:

We used a collection of Gal4 lines including *IR40a-Gal4*, *IR68a-Gal4*, *Nan-Gal4*, *IR31a-Gal4* and *IR41a-Gal4*. All of these lines have been reported to label neuron groups that are responsive to humidity changes or water stimulus (PMID: 21940430, PMID: 17994098, PMID: 28621663, PMID: 28736172, PMID: 27656904, PMID: 27161501). We found only two lines (*Nan-Gal4* and *IR68a-Gal4*) showed obvious defect in moisture attraction when crossed with the UAS-ReaperHid line (Fig. 1g). It is not surprising that moisture attraction requires *IR68a-Gal4* labeled moist cells that mediate moisture sensing in an IR (ionotropic receptor)-dependent manner. We further identified *Or42b-Gal4* positive neurons as a subset of *Nan-Gal4* neurons that are required for this behavior (Fig. 1g and Supplementary Fig. 2a). Moreover, when simultaneously ablating Or42b neurons and IR68a neurons, we found the 70% RH-induced attraction can be entirely blocked (Fig. 1g).

The attraction indexes of *IR25a²* and *IR68a^{MB05565}* flies were reduced (albeit not abolished) to similar levels (Fig.4f) in a 20% to 70% RH gradient, consistent with the result that dry cells labeled by *IR40a-Gal4* were dispensable for this behavior. When changing the humidity setting to a 70% to 96% RH gradient, *IR25a²* flies showed a humidity-blind phenotype (Supplementary Fig. 3d), which coincides with previous work (see Fig. 4 in PMID: 28621663) and suggests our assay is reliable. Following the reviewer's suggestion, we have included all the corresponding data in the revised

manuscript.

o The flies used in the assay have been starved and desiccated. Since Or42b neurons are associated with food search it would be of interest to see how sated behave in the arena? Or starved but not desiccated (Grown on pure agar a couple hours)? How does that influence the phenotype of Or42b-Gal4>ReaperHid/Kir (only starved) ?

Response:

We tested sated flies and starved flies (grown on pure agar overnight) using our behavioral assay, they behaved like antenna ablated flies that are humidity-blind (Supplementary Fig. 1d). These observations reflect the significance of internal state in driving the moisture seeking behavior, consistent with previous results (PMID: 14910691, PMID: 25262493). Since a single assay was finished within 2 min, we did not observe a clear preference for 70% RH in sated flies as reported before (PMID: 27161501).

As starved flies did not show obvious spatial distribution in our behavioral tests, we were unable to tell how starvation affects the phenotype of Or42b-Gal4>ReaperHid/Kir flies.

o Figure 1E error bars are not visible

Response:

Thanks for the suggestion. We have made corresponding corrections (Fig. 1e).

• Figure 2:

o The presented mechanism for humidity-induced structural changes in olfactory sensilla strikes me as unlikely. ab1 sensilla are exposed on the surface of the antenna and a gust of wind would deflect the sensillum to a similar degree as the suggested humidity-induced changes. How would the fly differentiate between wind and humidity? Hygrosensilla in *D. melanogaster* and in other insects are located in positions where they are shielded from air (eg in the sacculus) for this reason.

Response:

Deflection of bristle can modulate the activity of mechanosensory neurons by direct

contact between the external structures and the dendritic tip (PMID: 9438251). However, this process is unlikely to happen in ab1 sensilla due to the different morphological structures of mechano- and olfactory sensilla. To avoid misleading, we have removed the deflection data.

After improving the experimental conditions and methods according to the reviewer's suggestions, we have now shown that elevated moisture levels could alter the cuticular curvature of ab1 sensilla, which might result from the hygroscopically induced swelling of the cuticular wall. By using a constant dry airflow as control, we confirmed that a gust of wind itself is unable to elicit a similar shape change (Fig. 2). Moreover, wind sensing relies on subsets of Johnston's neurons in the 2nd antennal segment of *Drosophila* (PMID: 19279637), while humidity sensing neurons are located in the 3rd antennal segment. Wind and humidity are also separately coded in fly's brain. Such a mechanism allows flies to differentiate between wind and humidity.

Morphological studies proposed that hygrosensilla should be poreless sensilla located in an invagination to protect them from mechanical irritation or damage (PMID: 8565054) and the sacculus sensilla fit well into this notion. However, several studies have reported that basiconic and coeloconic sensilla located on the surface of *Drosophila* antenna also harbor neurons that are responsive to humidity, although they could not be classified as hygrosensilla (PMID: 16162917, PMID: 17994098, PMID: 21940430). These surface located sensilla hairs were surrounded by a large number of non-innervated spinules (PMID: 11395013), which could also play a shielding role to some extent.

o No statement about humidity levels are presented in this figure. At what RH and temperature where the experiments performed and how much was the humidity changed to induce the deflection?

Response:

Sensilla imaging experiments were performed at room temperature (25°C) and room humidity was controlled at 50-60% RH. We used a thin tube (~0.2 mm diameter) to provide dry/moist airstream in the sensilla imaging experiments, the humidity change

elicited under this situation was severely limited. We only achieved a humidity change from $46.64 \pm 0.39\%$ to $60.88 \pm 0.42\%$ RH, as verified by the Sensirion electronic hygrometer. This humidity increase already sufficed to induce the sensilla deformation (curvature change).

o Only dry -> moist stimulus was used. Adding moist -> dry would be easy to do and would give valuable data. It would be expected to return the change in deflection and curvature.

Response:

Thanks a lot for this constructive advice. We have modified the humidity stimulation to a dry -> moist -> dry stimulus and the control group was challenged with a dry -> dry -> dry stimulus. As expected, the cuticular curvature change was returned to the baseline level following the switch from moist air to dry air (Fig. 2).

o Do other sensory hairs, especially basiconic, change shape in a similar way? as ab1? As far as I know all sensilla are composed of similar material (chitin) that would interact with water vapour in similar ways as ab1. Does this mean all basiconics are responsive to humidity?

Response:

Thanks for this suggestion, we agree that hygroscopic swelling is likely to occur in other basiconic sensilla given the similar material (chitin) constituting their cuticular wall. Our data showed the curvature of Or42b>CD4-GFP negative large basiconics also changes upon exposure to humidity stimulation (Supplementary Fig. 5). A detailed analysis revealed a strong negative linear correlation between the baseline radius of curvature and cuticular deformation ratio (Fig. 2f), which suggests the curved basiconics are more responsive to humidity while straight basiconics may react poorly. Based on the above results, we think not all basiconic sensilla are responsive to humidity.

o The example images used in this figure look very blurry. How can you accurately measure such fine changes in deflection (only three degrees) and curvature from this material? I would recommend trying calcofluor or UAS-ChtVis-Tomato (PMID:

26395478) for better contrast. But for best contrast and analysis you would probably need to study these effects using electron microscopy?.

Response:

We have improved the experimental conditions and methods according to the reviewer's suggestions. Now the images with higher clarity and contrast are presented in the revised manuscript (Fig. 2), which we hope could more convincingly support our claims. Humidity changes from $46.64 \pm 0.39\%$ to $60.88 \pm 0.42\%$ RH could increase the radius of curvature of ab1 sensilla from $21.73 \pm 1.37 \mu\text{m}$ to $30.94 \pm 1.56 \mu\text{m}$, while switches from $46.64 \pm 0.39\%$ to $46.64 \pm 0.39\%$ RH had no such effects.

In mechanoreceptors, the dendritic tip of sensory cells contacts the external structures such as bristles, so that the deflection of bristles can modulate the activity of mechanosensory neurons (PMID: 9438251). However, this process is unlikely to happen in ab1 sensilla due to the different morphological structures of mechano- and olfactory sensilla. To avoid misleading, we have removed the deflection data.

Electron microscopy is currently out of our reach and it might be hard to determine the effects of humidity on sensilla structure after treatments required for electron microscopy. We have tried UAS-ChtVis-Tomato, but the photobleaching of fluorescence signal is fast, which hinders further imaging experiments. We hope the data presented in the revised manuscript could well support our conclusion.

o More details about how the measurement of deflection and curvature should be included in the methods section. How was deflection angle calculated?

Response:

The curvature analysis is conducted using the Kappa plugin of Fiji according to the documentation on Github (<https://github.com/brouhardlab/Kappa>). We first aligned the image stack using the StackReg plugin of Fiji, only the image stacks in which the sensilla base displayed no movement were opened in Kappa for the next analysis. In Kappa plugin, 5 points that track the midline of the sensilla were first manually defined to make an initial curve. We then adjusted parameters such as the colour channel, the brightness threshold, and the distance from the curve to make all GFP signals or bright

pixels within the sensilla contour selected. Next, we fitted the initial curve to the chosen pixels by the least square algorithm in Kappa. After setting the scale factor ($\mu\text{m}/\text{pixel}$) of the images, we exported the average radius of curvature of the fitted curve for statistical analysis. This detailed information has been added to the method section.

In the original manuscript, to calculate the deflection angle of sensilla, we set the sensilla base as the vertex of angle and connected the base and apex of the sensilla to make the edge. The Fiji angle tool was used to calculate the deflection angle. The deflection data have been removed to avoid misleading as explained above.

- **Figure 3:**

- o TMEM63 expression is showed for a thin section of the antennal lobe. Which other glomeruli are TMEM63 expressed in? It looks in the image as the V-glomerulus is labelled. Assuming a humidity-induced change in shape of ab1 is trasduced by TMEM63 expressed in ab1A-neurons, a similar response would be seen in ab1C-neurons if they also express TMEM63. Are ab1C also responsive to humidity?

Response:

Thanks for this important question. TMEM63 is expressed in multiple neuronal groups in the antenna. Although we were not able to examine the expression profile of TMEM63 in each of the ~50 olfactory sensory neuron groups, we have shown that TMEM63 is expressed in both OR (olfactory receptors)-expressing cells and IR-expressing neuron groups (Supplementary Fig. 10a).

As Reviewer #1 noted, $Tmem63^{LexA}$ labels the V glomerulus innervated by ab1C neuron axons (Supplementary Fig. 10a). The ab1 sensilla harbors four olfactory sensory neurons including Or42b neurons and Gr21a neurons (ab1C neurons), thus a curvature change in the sensilla cuticle seems to activate both of them. We have conducted calcium imaging with the $Tmem63^{LexA}$ driver but did not observe a humidity response of the V glomerulus (Fig. 3c-h), which should be visible on the same focal plane as VP4 (PMID: 25327641, PMID: 26912260). Considering the mechanism of humidity transduction in the mechanical hygrometer model, TMEM63 might needs other components to constitute a hygrometereceptor. Another possibility is that TMEM63 was not

targeted to the sensory ending of Gr21a neurons.

o Is TMEM63 expressed in any of the known humidity-responsive structures the sacculus/VP-glomeruli?

Response:

By double-labeling experiments, we showed that TMEM63 is expressed in Ir40a neurons but not Ir68a neurons (Supplementary Fig. 10b-e). However, the physiological responses of Ir40a neurons to dry stimulation was independent of TMEM63 function (Supplementary Fig. 11).

o No error bars for control in 3f or 3g.

Response:

Thanks for the suggestion. We have made corresponding corrections (Fig. 4c, d).

• Figure 4:

o How was humidity stimulus delivered for the calcium imaging experiments? Since ab1 sensilla seems to be sensitive for deflection it would be important there is no change in flow rate when stimulus is changed. As the methods section is written it is not clear how the flow was generated, just that it was created by a syringe. What pressed the syringe? A flow rate of 500 ml/min is given, how, where and when was this measured?

Response:

By using constant dry airflows as control, we have shown that switch between air flows is unable to elicit a shape change of ab1 sensilla (Fig. 2). The humidity stimulus was generated by similar protocols from a previous study (PMID: 27656904) except that the syringes were manually pressed in turn by two persons. Although air flow was generated by manual operation but not by automatic vacuum pumps, the humidity responses were robust and stable across different humidity changing cycles and calcium responses only occurred at the time point of humidity switch (Fig. 3i).

The volume of the syringe used was 50 mL, hence 50 mL air was passed during the dry or moist stimulus period which was 6 s. From this protocol, we can calculate the

flow rate as 500 mL/min.

o As I wrote above, the ab1A neurons have been tested with a water stimulus before with no response and with a response to a putative dry stimulus (methanol/ethanol). How can these discrepancies be explained? The calcium imaging data in this paper shows that the Or42b neurons even respond more strongly to moisture than to apple cider vinegar. Does this mean humidity is the primary stimulus these neurons are responding to? What happens if you present apple cider vinegar at different humidities? Does it modulate the response on the Or42b neurons?

Response:

The discrepancies have been explained and discussed (see above) and these issues likely arise from misunderstanding of our data. We have showed the difference between water stimulus and the humidity changes in our work and we found evidence that methanol/ethanol cannot produce dry stimulus.

It should be declared that our data did not indicate Or42b neurons respond to humidity more strongly than to vinegar. We are sorry that the dilution ratio of odorants was not correctly labeled in the original manuscript (Fig. 5e, f), which might lead to misunderstanding. In our tests, a relatively low concentration of vinegar (10^{-3} dilution) and ethyl propionate (10^{-5} dilution) was used. The small response to such low concentration of vinegar (0.1% v/v) is in accord with previous studies (PMID: 23457557, PMID: 26439011).

There is no doubt that Or42b neurons are olfactory sensory neurons, but our data claims a multimodal role of these neurons. In addition to detecting food odors, they also partake in detecting specific humidity levels in thirsty flies. Considering the multimodal role of Or42b neurons, it would be interesting to see how olfactory response and humidity response influence each other in this neuron group. We thus tested the vinegar response when simultaneously applying the dry airflow. To our surprise, Or42b neurons responded more strongly to 0.1% vinegar in the dry condition (Fig. R1). Further investigations are required to elucidate the physiological significance of this phenomenon, which is beyond the scope of this study.

Fig. R1

o When activated by vinegar smells, Or42b-neurons signal approach behavior (PMID: 19396157) and here Or42b neurons are shown to be activated by humidity even more strongly than vinegar. Therefore, a humid stimulus would generate a stronger approach behavior than apple cider vinegar. A suggested experiment would be the humidity behavior arena from Figure 1a with moisture in one test well and apple cider vinegar in the other test well. Will the flies go for moisture (as this data indicates) or apple cider vinegar. The experiments can be performed in *Ir25a2* flies where the sacculus neurons are not functioning, so they won't impact the outcome.

Response:

To quantitatively compare the humidity response of Or42b neurons to the olfactory response of Or42b neurons, we have acquired more calcium imaging data for higher chemical concentrations. We found Or42b neurons in wild-type flies and *Tmem63^{KO}* mutants still showed similar calcium responses to apple cider vinegar at concentrations of 1% and 10%. The response to 10% vinegar (wild-type: $60.18 \pm 3.34\% \Delta F/F_0$) was comparable to the humidity response (wild-type: $56.76 \pm 3.89\% \Delta F/F_0$) in our previous data (Fig. 5c, f). However, the F_0 level of humidity response was calculated in the dry condition. Given the enhancement of olfactory response by dry airflows (Fig. R1), it is likely that the response to 10% vinegar should be far larger than the humidity response when setting to the same baseline F_0 level.

In addition, it should be noted that flies in our behavioral tests were unlikely to experience a sharp humidity change (~25% to ~67% RH within 1 s) (Supplementary Fig. 1a), especially given that they locomoted in the 60 mm diameter arena with a speed of ~9 mm/s (Supplementary Fig. 7d). Consequently, the different time scales of functional imaging and behavioral assay make it not feasible to validate the imaging results by behavioral tests. Furthermore, a recent study showed that “thirsty flies seek water over food, and hungry flies seek food over water” (PMID: 34018925), so whether the flies go for moisture or apple cider vinegar when given a choice depends on the deprivation state.

o Error bars are missing from the traces in 4b

Response:

The traces in Fig. 5b and 5e are representative traces from single experiments, not mean \pm SEM.

• Figure 5:

o The humidity measurement in 5b does not seem to represent experimental conditions. In the experiment five breaths are given before the hand is presented. How do these breaths change the humidity? I assume humidity (and temperature) would be increased? A better way of performing these experiments would be to have a constant stream of CO₂ to activate search behavior instead of giving breaths.

Response:

Thanks for the suggestion. We agree that using a constant stream of CO₂ to activate search behavior would yield better results. As five breaths were not applied to the whole testing box rather than the upper space of the presented hand, the humidity and temperature at different positions in the box should be equally affected. Moreover, the presented hand was positioned in different areas from trial to trial. We therefore think the current form of data is sufficient to support the idea that *Tmem63* knockdown influences the host approach.

o The effect seen in 5d is very small. Based on this, can you really state that host approach in mosquitos “requires Tmem63”? It seems like an overstatement to me. The

mosquito part of this manuscript deserves its own separate study to be able to say anything about mosquito hygrosensation.

Response:

There might be two reasons that can explain reduced but still robust attraction behavior for hand. First, odor, CO₂ and thermal cues can still guide mosquitos to approach the hand; second, the partial knockdown may leave residual TMEM63 to influence the behavior. Moreover, the deficiency observed at the 20 s time point is statistically significant (Control: $52.0 \pm 2.2\%$; *Tmem63* knockdown: $29.3 \pm 2.2\%$). We hope the mosquito data could support the idea that TMEM63 plays an evolutionarily conserved role in humidity sensing. We have changed the statement “require TMEM63” to more accurate interpretations. If the reviewer thinks it is necessary, we can remove mosquito data from the manuscript.

Reviewer #2 (Remarks to the Author):

The study by Li et al. reports on the identification of a subpopulation of olfactory sensory neurons (Or42b) as mediators of hygrosensation using *Drosophila* as a model system. The study addresses an open question on how hygrosensors transduce humidity signals into electrical signals that drive behavioral responses by implicating mechanical deformations of the cuticular sensilla housing Or42b neurons via the action of the mechanosensitive ion channel TMEM63 within these neurons. The authors document evolutionary conservation of TMEM63 orthologues via cross-species rescue experiments with human TMEM63B and present evidence for a role of mosquito TMEM63 in host-seeking via humidity cues.

A major strength of this manuscript is the insight into the mechanisms by which humidity signals are transformed into electrical activity to drive stimulus-relevant behavior by providing evidence in support of a mechanical hygrometer model via the action of TMEM63. Overall, the work is largely compelling, carefully performed and should prove of interest to others in the field. The development of a novel behavioral assay should also facilitate future studies to glean additional insights in the molecular control of this process coupled with potential insights into putative targets for disease

vectors that utilize humidity cues.

The statistical analyses and descriptions of the methodologies and approach are clear and supported.

Response:

We greatly appreciate the reviewer for the careful reading and very positive and supportive evaluation “*Overall, the work is largely compelling, carefully performed and should prove of interest to others in the field. The development of a novel behavioral assay should also facilitate future studies to glean additional insights in the molecular control of this process coupled with potential insights into putative targets for disease vectors that utilize humidity cues.*” We are also grateful to the reviewer for nice summary of our work and for the constructive suggestions to improve the study.

Following the reviewer’s suggestions, we have made corrections in the revised manuscript. Additional data have been included in the revised manuscript and please see our specific responses to the reviewer’s comments below.

Below I summarize my primary critiques/comments on the manuscript that the authors should address:

Critiques/Comments:

Page 3, line 13: The authors state that wild type flies showed robust moisture attraction within 90 seconds. This is certainly true, though the data in Fig. 1C show that attraction response appears to peak much earlier at 50 sec.

Response:

We apologize for this inaccurate description. Wild type flies showed robust moisture attraction within 50 seconds. We have revised this sentence as the reviewer suggested. Since several tested groups showed a delayed attraction, we only used the attraction indices at plateau stage for quantification analysis.

Page 4, line 12-14: What is the nature of the Or42b[EY] allele with respect to disruption of this gene? Can the authors exclude a role of olfactory receptors entirely in

hygrosensation as the sentence reads vs. editing the sentence to reflect that the data suggest that Or42b is not involved in hygrosensation pending clarification on the nature of this EY allele.

Response:

The Or42b[EY] allele is a P element insertion in the second exon and caused defects in behavioral response to low concentrations of ethyl acetate, suggesting the disruption of this gene (PMID: 18614033). Another mutant line tested, Orco2, was completely normal in humidity induced attraction (Fig. 3a). This line has been reported to block the function of other olfactory receptors (including Or42b) and is olfactory-blind (PMID: 15339651). We have revised the sentence to “indicating that ORs in Or42b neurons are not involved in hygroreception”.

Page 4, Lines 26-28 and Supplementary Figure S3: The authors show that *iav* mutants perform normally in their assay. How do the authors square these results with previous studies showing that *iav* mutation affects hygrosensation? (Liu L et al, 2007. doi: 10.1038/nature06223). This should at least be addressed in the text.

Response:

Thanks for pointing out this important issue. In the Nature paper, Liu et al. conducted a classical T-maze assay in which water sated flies were introduced to choose from 0% RH air and 100% RH air. However, in our study, we established a modified hygrotactic behavior assay allowing desiccated flies to respond to a 20-70% RH humidity gradient. While *iav* participates in the humidity choice between 0% and 100% RH in water sated flies, it is not required for humidity attraction to 70% RH in desiccated flies. The differences in humidity settings and internal state of animals might lead to distinct results. We have also revised the corresponding part in the manuscript.

Page 6, Line 26: The authors should specify the odorants in the results text. Also, the statement that it is “dispensable for odor detection” is too broad. If the authors think the use of apple cider vinegar and ethyl propionate support a broad interpretation, they should specify why, otherwise the statement should be modified to reflect that this is only true for the odorants specifically tested.

Response:

Thanks for this suggestion. We have corrected this statement to “dispensable for the detection of vinegar and ethyl propionate”.

Page 23, line 10: I am not entirely clear on what the authors mean by “results from one of three independent experiments are shown” given that the data in Fig. 5a display SEM bars?

Response:

We are sorry for this vague description. For a independent experiment, we retracted RNA from 10 mosquitoes in each group (control and dsTMEM63), then the relative expression level of Tmem63 from each animal was calculated for statistical analysis. We have revised the figure legend to avoid confusion.

Fig. 1: What is the efficiency of the Or42b ablation(GFP/UAS-Redstinger) using ReaperHid? Can the authors show data demonstrating this in combination of Or42b labeled neurons?

Response:

According to the review’s suggestion, we have performed neuronal ablation in Or42b>RedStinger flies. The efficiency of neuronal ablation is nearly 100% (Supplementary Fig. 2b).

Fig. 2: What is the time scale for the change in curvature observed upon exposure from dry-to-moist humidity? How long does the mechanical deformation last? In Fig. 2c, how do the authors explain the difference in baseline radius of curvature for dry conditions given that the significant increase in curvature observed with dry-to-moist appears due to the fact that the baseline dry average curvature is lower?

Response:

Thanks for this valuable input. According to the reviewer’s suggestions, we have improved the experimental conditions and methods. To test how long the moisture-induced curvature change can last, we applied a moist airflow for more than 10 sec. The radius of curvature of basiconic sensilla increased to the peak level within 0.992 sec after the humidity switch from $46.64 \pm 0.39\%$ to $60.88 \pm 0.42\%$ RH, which matches the

time scale of humidity responses in calcium imaging. This mechanical deformation last throughout the entire period of moist airflow (Supplementary Fig. 5f).

We are sorry that the baseline radius of curvature for dry conditions was not well controlled. We have conducted the experiment again with several modifications and replace the original results with new ones with a similar starting point (Fig. 2).

Fig. 4: For the GCaMP6 experiments performed in Fig. 4, why do the authors switch from using Or42b-GAL4 to using Nan-GAL4? Can the authors provide data to show the overlap between these two drivers? In the Fig. 4 legend, the authors refer to the calcium responses in Or42b neurons, but do not sufficiently establish that Nan-GAL4 labels the same population of neurons. Also, in the Fig. 4 legend, the genotypes do not include the RFP normalization transgene.

Response:

For rescue experiments, we need a recombination chromosome to get the flies carrying the homozygous *Tmem63^{KO}* (II) allele, transgenes *UAS-GCaMP6m*, *UAS-tdTomato*(III), *UAS-DmTMEM63*(III), and the Gal4 driver. One recombination line *Tmem63^{KO}, Nan-Gal4* was successfully generated, so we used this line for calcium imaging experiments.

By using a Gal80 line of Or42b as suppressor for Nan-Gal4, we found it indeed erased the GFP staining of neuron projections in glomeruli DM1, suggesting the overlap between the two drivers (Supplementary Fig. 2a).

We are sorry that the genotype was not correctly shown in the legend during the preparation of the original manuscript. As described above, a recombination line *UAS-GCaMP6m, UAS-tdTomato* with transgenes inserted on the third chromosome was used.

Supplementary Figure S3: Why did the authors choose different molecule subsets for the mutant and RNAi approaches?

Response:

For some candidate genes (*pzl* and *ppk*), the null mutant lines are not available, we thus tested RNAi lines instead. Since *nanchung* has been reported to play essential roles in thirst state sensor in central brain (PMID: 27477513), which can also affect the moisture

attraction behavior, we chose to perform RNAi-mediated knockdown of this gene in Or42b neurons. Furthermore, according to the results yielded by Or42b-Gal4-driven knockdown, we can attribute the phenotype to the contribution of the corresponding gene's function in Or42b neurons.

Reviewer #3 (Remarks to the Author):

In this study, Li et al. reports structural and molecular evidence for mechanosensitive molecules to act as hygrosensors and mediate hygrosensation in *Drosophila*. The Authors suggest that this evidence supports the mechanical hygrometer model, in which humidity changes can be transformed into mechanical cues, subsequently used by *Drosophila* to locate environments with preferred humidity levels. A secondary question involved investigation of whether a mechanosensitive molecular homologous in the mosquito vector is also involved in this insect's host seeking behaviour.

The Authors approaches a fundamental question in environmental sensory transduction which is of broad interest, yet still relatively little understood (i.e. when compared to thermosensation). The study involves sophisticated genetic, molecular, structural, and electrophysiological methods. Furthermore, some of the results are novel, and they certainly add to the understanding of the molecular architecture of hygrosensation in invertebrates. However, I have several concerns regarding the validity of the modified hygrotactic assay reported by Li et al., and its suitability (in current form) for providing the conclusive evidence the Authors ultimately rely upon for their claims.

Response:

We appreciate the reviewer's summary of our work and valuable comments. We have performed additional experiments to enhance the validity of our previous data and we believe the revised manuscript is significantly improved.

While the genetic, molecular, structural, and electrophysiological data are important mechanistically, the behavioural data are the primary outcomes here; hence, it is my opinion that the validity and reliability of the Authors' new assay is of paramount

importance to ensure that the mechanistic evidence can be confidently relied upon.

My first, main question assessing this study was to determine whether flies showed a sufficiently robust and (most importantly) a clear humidity-dependent-behaviour. Unfortunately, looking at some of the data reported (e.g. attraction indices under normal conditions no greater than 60% and under disrupted conditions no smaller than ~30%) I feel the latter may not be the case.

Response:

We agree with the reviewer that the reliability of our behavioral assay is of great importance to support the conclusion drawn in our work. Attraction indices of ~60% are broadly consistent with previous literature, some studies even reported that attraction index fell below 50% after three minutes (PMID: 25738801, PMID: 34018925). The chance level should be calculated from an uniform distribution (the area of the 20 mm diameter moist well divided by the area of the 60 mm diameter arena), which is ~12% but not 50%. Given this information, the 60% level should be considered a robust behavior. The aggregation of flies to the moist area is indeed distinguishable and robust in our tests (Fig. R2, see Supplementary Movie 1 for more details).

Fig. R2

We have further shown that a double mutant line for *Tmem63* and *Ir68a* reduced the moisture attraction to $13.9 \pm 1.2\%$ (Fig. 4f), which is close to the 12% random distribution. Both of the two molecules are found to be essential for physiological responses to moist air (PMID: 28736172, PMID: 28621663, our calcium imaging data

for Tmem63), suggesting that attraction of the flies to the 70% RH area shown in our paper is a humidity-dependent behavior.

Secondly, there are important information about the assay which are missing, i.e. what where the temperature conditions within this and where temperature gradients present? I feel this information is particularly important to better untangle the multisensory nature of this hygrotactic behaviour, that is, what is the complementary role of thermosensory cues in hygrosensing, as previously reported (see also next comment).

Response:

Thanks a lot for this advice. We have performed concurrent experiments to determine the complementary role of thermosensory neurons in this behavior (Fig. 1f, Supplementary Fig. 3e). Please see below (response to point #6) for more details.

Thirdly, and perhaps most importantly, it seems to me that the evidence reported indicates that while important, TMEM63 is not essential for hygrosensing, i.e. its absence does not completely abolish hygrosensing. Indeed, under Tmem63KO conditions, the attraction index went from ~60 to ~30% (Fig. 3), indicating that this molecular disruption only halved the likelihood of attraction (i.e. 30% of flies still gathered in the 70% RH, which is roughly three time as much as the 12% random distribution). Accordingly, I would have expected the authors to perform concurrent experiments to determine the complementary role of thermosensory molecules in the remaining hygrosensing function in their flies, in a way similar to the approach taken by Russell et al. with the C.elegans (see <https://www.ncbi.nlm.nih.gov/pmc/articles/PMC4050571/>). As the Authors also report when discussing the mosquito data, it may be likely that flies also integrate multisensory cues in hygrosensing (in a way similar to what humans do, see <https://pubmed.ncbi.nlm.nih.gov/24944222/>); hence, concurrent thermo- and mechano-sensory experiments could have shed a better light on the role of multiple hygrosensing molecules in this fly's behaviour.

Response:

We have found a number of studies reporting a partially reduced behavior phenotype

when disrupting the genes that function as the sensory mediator/transducer (e.g. PMID: 17450139, PMID: 22343891, PMID: 23222543, PMID: 27478019). Importantly, a prior study has also revealed that single mutation in either dryness sensing pathway (Ir40a) or moisture sensing pathway (Ir68a) is not sufficient to abolish the hygrotaxis of desiccated flies (Fig.R3). In this regard, our results are in accord with previous investigations. We think it is possible that other sensory pathways play a complementary role in the remaining hygrosensing function of *Tmem63^{KO}* flies.

Fig. R3 (Figure 4b from Knecht ZA et al., 2017)

Regarding the residual behavioral response of *Tmem63^{KO}* mutant flies, we have some different opinion with the view raised by reviewer #3. Although the mechano- and thermo- sensory integration could explain this partly diminished phenotype, the possibility that multiple hygrosensory inputs work together to guide the behavior should not be ruled out. As mentioned above, flies that are doubly mutant for *Tmem63* and *Ir68a* showed a disrupted moisture attraction. Moreover, concurrently blocking *Or42b* neurons and *Ir68a*-expressing neurons also led to a random distribution in the arena (Fig. 1g). We also found that *Tmem63* is not expressed in *Ir68a-Gal4*-labelled neurons (Supplementary Fig. 10c. e). These results indicates that *Ir68a* is responsible for the residual behavioral response of *Tmem63^{KO}* flies.

All in all, I believe that, while certainly providing interesting data, the main conclusions

of the current manuscript rely on experimental evidence that is not as strong as one would hope for. Accordingly, I would invite the Authors to consider these general comments, along with my list of detailed queries reported below.

Response:

Following the reviewer's suggestion, we have performed additional experiments and incorporated extra data in the revised manuscript, which we hope could improve the reliability and validity results and claims. Below are our point-to-point response to the reviewer's comments.

Detailed comments

1. Please introduce the model organism used in the title of the manuscript (i.e. "...mediate hygrosensation in *Drosophila*").

Response:

Thanks for this concern. As we felt *Tmem63* might fulfill an evolutionarily conserved role as a hygrosensor (e.g. in flies, mosquitoes and probably reptiles and birds), the model organism was not emphasized. We have revised the title as the reviewer suggested.

2. Please provide more information about the validity and reproducibility of this hygrosensing assay – what was the temperature gradient within the assay? Why did you opt for such a short duration assay (e.g. when compared to Enjin et al. where 4h were allowed, which resulted in attraction index of ~1)?

Response:

By neuronal ablation and mutant screening, we both identified a complementary role of IR68a in the residual behavioral response of *Tmem63*^{KO} mutants (Fig. 1g, Fig. 4f). When changing the humidity setting to a 70% to 96% RH gradient, *IR68a*^{MB05565} flies exhibited a reduced attraction to 96% RH while *IR25a*² flies showed a humidity-blind phenotype (Supplementary Fig. 3d), which coincides with previous findings (see Fig. 4 in PMID: 28621663). All the above information suggests that our hygrotactic assay is reliable and reproducible.

We measured the temperature gradient in the arena with a 20-70% RH setting using

the combination hygrometer/thermometer SHT31, but did not observe a temperature gradient within the arena (Supplementary Fig. 1b).

Since we set out to identify the molecules acting as the humidity transducer, which is regarded to respond to humidity changes within milliseconds, a short time course might be more accurate in reflecting the role of the hygrosensory pathway. Importantly, long duration assay with a group of flies might cause pheromone-related social cluster which makes the preference/attraction index not valid to evaluate the humidity sensing ability (PMID: 31959283). Moreover, a short duration assay is more suitable for functional screens.

3. Why only 60% of flies were attracted to higher humidity? This is just above chance level (50%), and it opens to the question of why ~40% of flies under desiccation are not attracted by 70% humidity?

Response:

As stated above, the chance level should be calculated from an uniform distribution (the area of the 20 mm diameter moist well divided by the area of the 60 mm diameter arena), which is ~12% but not 50%. Thus, an attraction index of 60% is much higher than the chance level.

Our assay as well as other food/water search assays allow the flies to freely walk in the arena, they show a searching behavior when encounter the odor/moisture source (PMID: 21458672, PMID: 25738801). For this reason a fraction of them were not staying above the moist well (Fig. R2). The attraction index is also dependent on the humidity levels. When changing the humidity setting to a 70%-96% RH gradient, wild type flies showed a higher attraction, with $78.3 \pm 1.8\%$ flies in the 96% RH region. However, *Tmem63* is dispensable for the hygrotaxis in this humidity range (Supplementary Fig. 3d).

4. What is the independent role of desiccation stress on fly's behaviour (independent of humidity-seeking behaviour)? Would it not be more appropriate to have a type of fly with a clear preference for that humidity level, and tests this with/without antenna to determine the true stress-free role of those antennae? After all, Enjin et al. 2016

showed that wildtype flies orient naturally (e.g. to 70%RH) based on their habitat of choice.

Response:

We have tested water sated flies using our behavioral assay, they behave like antenna ablated flies that were blind to the humidity gradient (Supplementary Fig. 1d). Since our study aims to explore the underpinnings of moisture attraction behavior in *Drosophila* and the internal state have been shown to be essential for this behavior (PMID: 14910691, PMID: 25262493), we cannot determine the true stress-free role of those antennae. Enjin et al. showed that wild type flies orient naturally to 70 % RH, however, this takes a quite long time period (more than 30 min). Long duration assay with a group of flies might cause pheromone-related social cluster which makes the preference/attraction index not valid to evaluate the humidity sensing ability (PMID: 31959283).

5. This assay gives only 60% attraction (which goes down to 15% when removing the antenna). However, when silencing odour neurons, attraction goes down to 36% (a drop of only 24%). It seems clear to me that there is something missing in the assay to drive full moisture attraction and that those odour neurons contribute to only half of that attractive behaviour (is this evidence of a thermo-mechano integratory system for moisture detection, as showed in humans?). This is a critical point as it challenges the Authors' conclusion that: Our study provides structural and molecular evidence supporting the mechanical hygrometer model, in which humidity changes can be transformed into mechanical cues so that mechanosensitive molecules may act as hygroreceptors to mediate hygrosensation. I would therefore invite the Authors to re-consider this claim.

Response:

We appreciate the idea of a thermo-mechano integratory mechanism for moisture detection, although it turns out that thermosensory pathway does not contribute to the moisture attraction in both the two humidity settings we used here (Fig. 1f, Supplementary Fig. 3e). Our behavioral data could not fit well into the mechanical

hygrometer model, but the results indicating humidity-dependent sensilla shape change together with calcium imaging data support that humidity changes can be transformed into mechanical cues and evoke the physiological response via the action of the mechanosensitive ion channel TMEM63. We have revised the related part in discussion to more accurately interpret our results.

6. Why not incorporating an approach like that of Enjin et al.'s to look at the thermosensory component too? Russel et al. in PNAS (<https://www.ncbi.nlm.nih.gov/pmc/articles/PMC4050571/>) took this approach with the c.elegans and provided elegant evidence for a multisensory integration mechanism. I would have expected the Authors to develop an experimental model where thermal cues are silenced (to confirm previous data); then mechanical cues are silenced (the current novel approach); and finally both cues are removed (in a way similar to the antenna removal). This experimental design would provide the most comprehensive and likely conclusive data on the fly's integratory mechanisms for hygrosensing. It is indeed important to note that, as also reported by Russell et al., moving across a humidity gradient can induce evaporative cooling which could provide thermosensory cues the animal may rely upon in their choice. This cannot be excluded in the current work, and may perhaps explain why half of KO flies retained humidity attraction?

Response:

Thanks for this valuable input. Despite the fact that no obvious temperature gradient is presented in our behavioral test, evaporative cooling can occur and provide thermosensory cues that guide animal behavior when flies are moving across a humidity gradient. When silencing all the six thermosensory neurons in the arista by the R11F02-Gal4 driver, the moisture attraction in both 20%-70% RH gradient and 70%-96% RH condition remained intact (Fig. 1f, Supplementary Fig. 3e). Then we concurrently blocked the six thermosensory neurons and Or42b neurons (mechanosensory pathway), the moisture attraction still persisted in these flies. Statistical analysis revealed that ablation of Or42b neurons alone has similar effects to concurrent ablation. All these data have been included in the revised manuscript to explore the complementary role

of thermosensory neurons in this behavior.

7. Fig. 2c – the radius of curvature (as a summary of the deformation induced by a switch to moist air) has a different starting point (i.e. dry baseline) between dry-dry and dry-moist, yet identical end points. Hence, while the 2 condition differ in their delta change, they have identical positioning under dry and moist conditions, which raises the question of whether humidity is actually being sensed via this mechanism? Why is the positioning different at baseline under the two dry conditions? I believe it is difficult to speculate that a hygroscopic movement has occurred in the dry.moist switch that was different to the dry.dry condition, (which the Authors use to justify the mechanical model). Furthermore, why are t-tests used for each comparison independently (dry.dry and dry.moist)? It would have been more appropriate to use a 2-way ANOVA with condition (i.e. dry-dry vs. dry-moist) and time (i.e. pre vs. post switch) as independent factors, to better ensure that baseline conditions would be equal prior to any change to sensilla cuticle positioning. It is my feeling that if this analysis were to be performed, no differences between conditions would be found. If that is the case, this is problematic, as it would further reduce the (already limited in my view) evidence that mechanical deformations in the cuticle have occurred under moist conditions and differently from dry conditions.

Response:

Thanks a lot for the advice on the experimental design and statistical methods. According to the reviewer's suggestion, we have conducted the experiment again and updated the original data. Now images with higher clarity and contrast are presented in the revised manuscript (Fig. 2), which we hope could more convincingly support our claims. Using a two-way ANOVA, we confirmed that the starting point of baseline radius of curvature were not significantly different for two groups (Fig. 2e).

8. Can you confirm all mutant and rescued experiments were conducted with similarly desiccated flies?

Response:

We found *Tmem63* is dispensable for the hygrotaxis in 70%-96% RH humidity gradient

(Supplementary Fig. 3d). The attraction index in this humidity range has been reported to rely on the degree of water loss (PMID: 25262493, PMID: 34018925). Thus, the attraction to 96% RH humidity can reflect the thirsty state of *Tmem63* mutant and rescued flies (Supplementary Fig. 12a). One way ANOVA followed by Dunnett's test confirmed that all these groups have no significant difference with the wild-type group, suggesting that these flies were similarly desiccated.

9. I appreciate an attempt to link some of the fly work to the relevance for a malaria vector. However, the section on the malaria vector feels somewhat disconnected from the whole paper. The evidence reported is not as thorough as that for the fly, and the findings are in my view, even less robust. Specifically, the mosquito data show only a 20% reduction in host approach with the mechano-channel KO. Furthermore, I also question the validity of this assay as the % of mosquito approaching the host is at the chance level (50%) at the 20s mark (the point used in the Discussion to exemplify differences). After the 20s mark, the mutant mosquito still shows a host approach, which in fact reaches higher levels than what the control mosquito reached at the 20s mark. My question is therefore: is this mosquito's host approach really disrupted to a meaningful extent? I doubt it, and I would therefore invite the Authors to consider whether reporting this data may oversell their general findings.

Response:

We hope the mosquito data could support the idea that TMEM63 plays an evolutionarily conserved role in humidity sensing. We have also revised the text to interpret these results more accurately. The chance level of the mosquitoes landing above the hand should not be 50% because mosquitoes fly rather than locomote like flies in the testing box. Considering that a human hand (15x5 cm) creates a humidity gradient in the 15 mm range above it and the testing box has a size of 15x15x5 cm, we estimate the chance level to be ~10%.

Humidity cues are believed to affect the landing and approach on human skin (PMID: 26394099). For this reason, hygro-sensory pathway works only at a short distance to the host, which makes hygro-sensory cues play more prominent role in the initial stage of

host seeking. We indeed observed a similar hand approach level for two groups after 3 min, but the deficiency observed at the 20 s time point is statistically significant (Control: $52.0 \pm 2.2\%$; *Tmem63* knockdown: $29.3 \pm 2.2\%$). There might be two reasons that can explain reduced but still robust attraction behavior for hand. First, odor, CO₂ and thermal cues can still guide mosquitos to approach the hand; second, the partial knockdown may leave residual TMEM63 to influence the behavior. If the reviewer thinks it is necessary, we can remove mosquito data from the manuscript.

10. There also very is limited information on the behavioural assay for the malaria vector, which hinders reproducibility (e.g. how many people where tested; how many repeats; etc.)

Response:

Only one person took part in the hand approach test for the malaria vector. The humidity gradient data was measured from the same person. We consider that only one participant would ensure the reproducibility of this assay, since the humidity gradient, temperature and odor emitted by different person might vary. For each independent behavior assay, we used 30-50 female mosquitoes. Seven and eight repeats were conducted for control and dsTMEM63 groups, respectively.

11. A primary point to consider in the Discussion is that the general evidence on mechano-integration for hygrosensing indicates a contribution of this mechanism at best, rather than strong evidence for the mechanical hygrometer model (as eventually claimed by the Authors).

Response:

According to the reviewer's suggestion, we have revise the related part in the Discussion. The statement is changed into "By neuron ablation methods and mutant analysis, we revealed a mechanosensory pathway that specifically contributes to 70% RH humidity-induced attraction. Our data also provide structural and molecular evidence supporting that humidity changes can be transformed into mechanical cues which evoke hygrosensory inputs via mechanosensitive molecules."

12. A secondary comment on the Discussion is that the argument here should be about

looking at how mechanosensory and thermosensory pathways contribute to hygrosensory strategies, rather than contrasting hygro- with thermosensory pathways (as the hygrosensory pathways may well rely on both mechano AND thermo pathways).

Response:

Since we have excluded a role of thermosensory pathways in the moisture attraction behavior, a multi-sensory integration strategy appears to be not the case for our study. However, we noted that such a thermo-mechano integratory system might work for the hygrosensors in the sacculus. Recently, a study demonstrated that a specific ionotropic receptor required for moisture detection, IR68a, is also a molecular receptor for warming cells in *Drosophila* larvae (PMID: 34452914). How IR68a plays a dual role in both thermosensation and humidity sensation is still mysterious. Further investigations on this molecule are required to answer this question. We hope the reviewer will find this provided information useful.

13. I note no sample size a priori analysis was used. Why? And importantly, why not using an a posteriori analysis to determine the power achieved given the effects size of interest? In this regard, what is the minimum relative change in hygrosensing behaviour that is sufficient to determine a critical role of a specific molecule in that pathway?

Response:

Thanks for the concern about the statistical methods. Power analysis was not used before experiments, we estimated the sample size based on previously published results on similar experiments. Following the reviewer's suggestion, we have performed post hoc analyses in PASS 15 (<https://www.ncss.com/software/pass/>) and confirmed statistical power > 0.8 for all the significant differences.

To estimate the minimum detectable difference, we assumed the common standard deviation to be 0.06, which is true for most groups. For RNAi or neuronal ablation experiments (two-tailed t test), we have power > 0.8 to detect a 0.125 attraction index change (Fig. R4). For mutant screening (one way ANOVA followed by Dunnett's test), we have power > 0.8 to detect a 0.2 attraction index change (Fig. R4).

14. Can the author provide accuracy values for their sensirion humidity sensors?

Response:

Of course, these data have been provided in the source data file accompanying the revised manuscript.

REVIEWER COMMENTS

Reviewer #1 (Remarks to the Author):

Li et al have made a significant effort to respond to the comments made by the reviewers. Many of the questions raised have been addressed satisfactory. However, some questions still remain, and new questions have arisen with the new data presented.

Specific points:

- New data is included using the R11F02-Gal4 crossed to UAS-Kir2.1. The authors use this line to rule out an effect of arista thermosensitive neuron on the observed phenotype. What the authors may not be aware of is that R11F02-Gal4 is also expressed in all sacculus neurons related to humidity (see glomeruli stained here https://flweb.janelia.org/cgi-bin/view_flew_imagery.cgi?line=R11F02, and if you stain antenna from R11f02>UAS:GFP you will see all sacculus-neurons stained). So what the authors are showing in Figure 1F is that humidity-sensing neurons and thermosensitive neurons are not required for attraction to the humid zone in their behavioral arena. In Figure 1G they go on to show that Ir68a-Gal4, labeling a subpopulation of R11F02-Gal4 neurons, is required for attraction. This math does not add up. I could imagine the UAS-Kir2.1 (used with R11F02-Gal4) is not as effective as UAS-ReaperHid (used with Ir68a-gal4). However, when crossed with Or42b-Gal4 UAS-Kir2.1 and UAS-ReaperHid give similar phenotypes. How can these discrepancies be explained?
- Multiple new glomeruli responding to moist and dry stimuli are reported. All express TMEM63 yet you get different responses and requirement of TMEM63 from these populations. DM1, response to moisture and express and requires TMEM63 for response; DL2 and DC3, response to moisture and express but do not require TMEM63 for response; VP4 and DP11, response to dryness and express but do not require TMEM63 for response; V, no response to humidity and express TMEM63. Is it likely that TMEM63 is a primary sensor to (a proxy of) moisture given these different observations? Should not all populations respond to moisture the same way and all require TMEM63?
- The experiments presented in figure 2 are now clearer to understand but I realize I still lack information. How long are the sequences at different humidity? At which time-point do you acquire the images? You use a confocal microscope and have GFP expressed in the sensilla you want to study, yet you use brightfield images for analysis. Why is that? The sensilla is a three-dimensional structure and a fast scan would give you that structure in all dimensions and a brightfield image will project this information onto two dimensions.
- TMEM63 is throughout the manuscript described as a mechanosensor. What is the argument TMEM63 is acting as a mechanosensor rather than an osmosensor? The sensilla studied all have pores, meaning water molecules could and likely do enter into the sensilla. Could TMEM63 equally well respond to changes in the osmotic concentration?
- The results in figure 6 are still lacking in rigor and I recommend removing it.

Reviewer #2 (Remarks to the Author):

I thank the authors for their careful attention to the original critiques and for adding new experimental data and modifying text to address those critiques.

My major and minor concerns have been addressed. I recommend publication.

Reviewer #4 (Remarks to the Author):

The authors have addressed all my comments satisfactorily, and I have no further recommendations.

RESPONSE TO REVIEWER COMMENTS

Reviewer #1 (Remarks to the Author):

Li et al have made a significant effort to respond to the comments made by the reviewers. Many of the questions raised have been addressed satisfactory. However, some questions still remain, and new questions have arisen with the new data presented.

RESPONSE:

Thank the reviewer for the positive comments on our revision. We have solved these new problems through supplementary experiments.

Specific points:

- New data is included using the R11F02-Gal4 crossed to UAS-Kir2.1. The authors use this line to rule out an effect of arista thermosensitive neuron on the observed phenotype. What the authors may not be aware of is that R11F02-Gal4 is also expressed in all sacculus neurons related to humidity (see glomeruli stained here https://flweb.janelia.org/cgi-bin/view_flew_imagery.cgi?line=R11F02, and if you stain antenna from R11f02>UAS:GFP you will see all sacculus-neurons stained). So what the authors are showing in Figure 1F is that humidity-sensing neurons and thermosensitive neurons are not required for attraction to the humid zone in their behavioral arena. In Figure 1G they go on to show that Ir68a-Gal4, labeling a subpopulation of R11F02-Gal4 neurons, is required for attraction. This math does not add up. I could imagine the UAS-Kir2.1 (used with R11F02-Gal4) is not as effective as UAS-ReaperHid (used with Ir68a-gal4). However, when crossed with Or42b-Gal4 UAS-Kir2.1 and UAS-ReaperHid give similar phenotypes. How can these discrepancies be explained?

RESPONSE:

Thanks for this concern about the R11F02-Gal4 line. The pictures cited here (https://flweb.janelia.org/cgi-bin/view_flew_imagery.cgi?line=R11F02) cannot show clearly that Ir68a-Gal4 neurons are a subgroup of R11F02-Gal4 neurons. To solve this problem, we checked the labeling of R11F02-Gal4 and R11F02-Gal4+Ir68a-Gal4 in the antennal lobe (Figure R1). We found that the neurons labeled by R11F02-Gal4 alone had almost no projection in the glomerulus VP5, where the axon termini of Ir68a-Gal4 positive neurons reside (Figure R2), so we got different results when blocking the neuron groups labeled by these two lines..

Figure R1

Figure R2 (Figure 2 M, N from Frank et al., 2017, PMID: 28736172)

- Multiple new glomeruli responding to moist and dry stimuli are reported. All express TMEM63 yet you get different responses and requirement of TMEM63 from these populations. DM1, response to moisture and express and requires TMEM63 for response; DL2 and DC3, response to moisture and express but do not require TMEM63 for response; VP4 and DP1I, response to dryness and express but do not require TMEM63 for response; V, no response to humidity

and express TMEM63. Is it likely that TMEM63 is a primary sensor to (a proxy of) moisture given these different observations? Should not all populations respond to moisture the same way and all require TMEM63?

RESPONSE:

Tmem63 is expressed broadly in antennal neurons, including the neurons targeting DM1, DL2, DC4, DP11, VP4 and V glomeruli. The “DC3” in the reviewer’s comments might be a typo. We have added data and showed that the humidity responsiveness of DC4 is independent of *Tmem63* (Figure R3), we also incorporated this data into the latest manuscript.

Figure R3

We have proposed that the curvature change in basiconic sensilla generates the force that activates TMEM63. However, the neurons projecting to DL2, DC4, DP11 and VP4 are housed in coeloconic sensilla (Silbering et al., 2011, PMID: 21940430; Shanbhag et al., 1995, PMID: 8565054), which are dramatically different from basiconic sensilla in the morphology (Figure R4). Curvature change is unlikely to happen in this type of sensilla and maybe for this reason TMEM63 cannot act as a primary sensor in these populations.

V glomeruli-targeting neurons, which are labeled by *Gr63a-Gal4*, indeed express *Tmem63*, but are not sensitive to humidity. It is speculated that it is related to the

difference of protein subcellular localization and distribution, we supplemented experiments and found TMEM63 is not targeted to the sensory ending of V glomeruli-targeting neurons (Figure R5).

Figure R4 (from Vosshall, 2000, PMID: 10981620)

Figure R5. Subcellular localization of TMEM63 revealed by crossing the *UAS-TMEM63::EGFP* transgene with *Or42b-Gal4* or *Gr63a-Gal4*. The *UAS-TMEM63::EGFP* construct harbors an EGFP tag in-frame fused at the C-terminus of TMEM63.

These observations do not contradict with our conclusion that TMEM63 is likely a primary sensor for humidity-dependent mechanical deformation. The scenario that cells expressing certain molecular receptors do not necessarily respond to related stimuli is often reported. For example, some DRG neurons expressing PIEZO have no mechanical sensitive response (Parpaite et al., 2021, PMID: 34731626). Labial GRN neurons expressing TMEM63 were also not related to the perception of food grittiness (Li et al., 2021, PMID: 33657409). Moreover, a variety of mechanosensitive molecules can be expressed simultaneously in the same group of neurons, but only a certain type of molecule is selectively required for sensory transduction. For example, chordotonal neurons in *Drosophila* and cochlear hair cells in mammals express PIEZO, a bona fide mechanotransducer, but they do not depend on the PIEZO for auditory transduction (Zhang et al, 2013, PMID: 23898199; Wu et al, 2017, PMID: 27893727).

- The experiments presented in figure 2 are now clearer to understand but I realize I still lack information. How long are the sequences at different humidity? At which time-point do you acquire the images? You use a confocal microscope and have GFP expressed in the sensilla you want to study, yet you use brightfield images for analysis. Why is that? The sensilla is a three-dimensional structure and a fast scan would give you that structure in all dimensions and a brightfield image will project this information onto two dimensions.

RESPONSE:

The experimental details are supplemented in the method part, and we have revised Figure 2 to indicate the time point of image acquisition. Images were obtained at a frame rate of ~2 Hz. For “dry-moist-dry” stimulation, we applied dry air for 8 s, then switched it into 5-s moist airflow and 8-s dry air in sequence. The time point in which the dry air was converted to moist air was defined as 0 s. We chose the images at time points -5 s, 2.5 s and 10 s for the analysis.

We have scanned the z-stack GFP signal and tried many times to capture three-dimensional structural changes of basiconic sensilla upon humidity changes. However,

humidity transduction occurs very quickly (within 1s), the result is lack of time resolution. The axial resolution of confocal microscope has a limit of $\sim 0.5 \mu\text{m}$ (Fouquet et al, 2015, PMID: 25822785), which is close to the thickness of sensilla we observed, so the results of fast scan and brightfield image will be similar.

Moreover, the GFP signal is not as clear as the bright field image, and the boundary of sensilla becomes indistinguishable compared with in the bright field. This is the reason why we used brightfield images for analysis.

- TMEM63 is throughout the manuscript described as a mechanosensor. What is the argument TMEM63 is acting as a mechanosensor rather than an osmosensor? The sensilla studied all have pores, meaning water molecules could and likely do enter into the sensilla. Could TMEM63 equally well respond to changes in the osmotic concentration?

RESPONSE:

Thanks a lot for this suggestion. We expressed *Drosophila* TMEM63 in HEK293T cells and performed calcium imaging. The results showed that cells expressing TMEM63 are more sensitive to osmotic pressure changes (Figure R6). Furthermore, expression of human TMEM63B restores the humidity-guided behavior in *Tmem63* mutants, indicating functional conservation. TMEM63B in mice have also been shown to be osmolarity sensitive in cultured cells (Du et al., 2020, PMID: 32375046). So it could not be ruled out that TMEM63 could mediate humidity perception by sensing osmotic pressure changes.

The data presented in our manuscript convincingly support TMEM63 as a mechanical force receptor to sense humidity-dependent sensilla deformation and mediate humidity perception. However, it is technically difficult to measure the changes in the osmotic concentration in the sensilla lymph, which is currently out of our reach. We have revised the discussion to make our conclusion more rigorous, our future work will continue to explore this possibility in depth.

Figure R6

- The results in figure 6 are still lacking in rigor and I recommend removing it.

RESPONSE:

We have removed this part from our manuscript.

Reviewer #2 (Remarks to the Author):

I thank the authors for their careful attention to the original critiques and for adding new experimental data and modifying text to address those critiques.

My major and minor concerns have been addressed. I recommend publication.

RESPONSE:

Thank the reviewer for the kind help in revising our paper and we are glad that the reviewer recommend publication.

Reviewer #4 (Remarks to the Author):

The authors have addressed all my comments satisfactorily, and I have no further recommendations.

RESPONSE:

Thank the reviewer for the approval of our revision.

REVIEWERS' COMMENTS

Reviewer #1 (Remarks to the Author):

I thank the authors for their patience with my comments. All comments have been answered satisfactory.

RESPONSE TO REVIEWER COMMENTS

Reviewer #1 (Remarks to the Author):

I thank the authors for their patience with my comments. All comments have been answered satisfactory.

RESPONSE:

Thank the reviewer for the time and effort into reviewing our manuscript and the highly valuable suggestions and comments.